



# Optimal closed-loop wake steering, Part 1: Conventionally neutral atmospheric boundary layer conditions

Michael F. Howland[1], Aditya S. Ghate[2], Sanjiva K. Lele[1,2], and John O. Dabiri[3,4]

[1]Department of Mechanical Engineering, Stanford University, Stanford, CA 94305
[2]Department of Astronautics and Aeronautics, Stanford University, Stanford, CA 94305
[3]Graduate Aerospace Laboratories of the California Institute of Technology (GALCIT), California Institute of Technology, Pasadena, CA 91125
[4]Department of Mechanical and Civil Engineering, California Institute of Technology, Pasadena, CA 91125

**Correspondence:** John O. Dabiri (jodabiri@caltech.edu)

**Abstract.** Strategies for wake loss mitigation through the use of dynamic closed-loop wake steering are investigated using large eddy simulations of conventionally neutral atmospheric boundary layer conditions, where the neutral boundary layer is capped by an inversion and a stable free atmosphere. The closed-loop controller synthesized in this study consists of a physics-based lifting line wake model combined with a data-driven Ensemble Kalman filter state estimation technique to calibrate the
wake model as a function of time in a generalized transient atmospheric flow environment. Computationally efficient gradient ascent yaw misalignment selection along with efficient state estimation enables the dynamic yaw calculation for real-time wind farm control. The wake steering controller is tested in a six turbine array embedded in a quasi-stationary conventionally neutral flow with geostrophic forcing and Coriolis effects included. The controller increases power production compared to baseline, greedy, yaw-aligned control although the magnitude of success of the controller depends on the state estimation
architecture and the wind farm layout. The influence of the model for the coefficient of power $C_p$ as a function of the yaw misalignment is characterized. Errors in estimation of the power reduction as a function of yaw misalignment are shown to result in yaw steering configurations that under-perform the baseline yaw aligned configuration. Overestimating the power reduction due to yaw misalignment leads to increased power over greedy operation while underestimating the power reduction leads to decreased power, and therefore, in an application where the influence of yaw misalignment on $C_p$ is unknown, a
conservative estimate should be taken. Sensitivity analyses on the controller architecture, coefficient of power model, wind farm layout, and atmospheric boundary layer state are performed to assess benefits and trade-offs in the design of a wake steering controller for utility-scale application. The physics-based wake model with data assimilation predicts the power production in yaw misalignment with a mean absolute error over the turbines in the farm of $0.02 P_1$, with $P_1$ as the power of the leading turbine at the farm, whereas a physics-based wake model with wake spreading based on an empirical turbulence intensity
relationship leads to a mean absolute error of $0.11 P_1$.



## 1 Introduction

Wind and solar energy are likely the only low-carbon energy technologies which are being implemented rapidly enough to mitigate the effects of global warming (IEA, 2017). Since wind energy has a low marginal cost (Bitar et al., 2012), increases in wind farm power production approximately manifest as a reduction in the levelized cost of electricity (LCOE) (Joskow, 2011). While the LCOE of wind energy is already often below those of traditional combined-cycle natural gas and coal (Bilgili et al., 2015; EIA, 2018), continued reductions in wind energy LCOE will likely increase the adoption of this technology (Borenstein, 2012; Ouyang and Lin, 2014) due to improved economics in sub-optimal wind resource areas (Wiser et al., 2015). Modern horizontal axis wind turbines achieve performance approaching the Betz limit (Wiser et al., 2015). However, collections of wind turbines arranged in wind farms suffer from aerodynamic interactions which reduce wind farm power production between $10$ and $20\%$ (Barthelmie et al., 2009) due to greedy control schemes which only consider the power maximization of individual wind turbines at the farm. Recent work has focused on the operation of wind turbines in a collective fashion in order to increase the power production of the wind farm through the mitigation of wake interactions (see review by Boersma et al., 2017).

Wind farm power optimization through wake interaction mitigation methods have generally relied on axial induction and yaw misalignment control since these two methodologies do not require significant hardware modifications on traditional horizontal axis wind turbines (Burton et al., 2011). Annoni et al. (2016) utilized a steady-state model to inform the optimal axial induction factors for each wind turbine in an LES of a model wind farm but did not find significant power production improvements over baseline greedy operation. Campagnolo et al. (2016a) found similar results in a wind tunnel experiment of three wind turbines. The full large eddy simulation (LES) adjoint equations were used to optimize the power production by Goit and Meyers (2015). Munters and Meyers (2017) and Munters and Meyers (2018) extended the work of Goit and Meyers (2015) and used dynamic axial induction and yaw misalignment to increase wind farm power production using the full-state adjoint. While these studies achieved successful dynamic power production increases over baseline operation, the computational expense of adjoint LES is similar to standard LES and is currently a challenge to use in real-time wind farm control. Bauweraerts and Meyers (2019) showed that coarse LES can potentially be used for real-time prediction and control but this requires future investigation and is not the focus of the present study. Ciri et al. (2017) used a model-free formulation and dynamic control to increase the power production of a model wind farm and found that downstream wind turbines may also need to change their operational strategy to increase farm performance. Gebraad et al. (2013) also used a model-free gradient-based optimizer to increase the power production of a wind farm by approximately $1\%$. Park and Law (2016) used data-driven Bayesian ascent to efficiently maximize the power production of a model wind farm using axial induction. As a general conclusion of the present literature, simulations and field experiments of stationary axial induction based on static-wake models (models which represent the time averaged behavior of a stationary or quasi-stationary flow) have shown this methodology is unlikely to increase wind farm power production in utility-scale wind farms but may improve wind turbine fatigue loading provided that the objective function is carefully proposed and the fatigue is accurately modeled (see extensive review by Knudsen et al., 2015). Meanwhile, dynamic axial induction or more sophisticated dynamic blade pitch strategies





may significantly increase power production (Munters and Meyers, 2018; Frederik et al., 2019, 2020) and require future field experimentation.

Greedy wind turbine operation minimizes the yaw misalignment between the nacelle position and the incoming wind direction. Contemporary wind turbines often operate in small yaw misalignment due to sensor noise and uncertainty (Fleming et al., 2014) leading to sub-optimal power production for the misaligned turbine. However, recent attention has focused on wake steering, the intentional misalignment of certain turbines within a wind farm in order to deflect wakes laterally away from downwind generators (Grant et al., 1997; Jiménez et al., 2010). While the yaw misaligned wind turbine's power production is decreased (Burton et al., 2011; Medici, 2005), wake steering has been shown to increase the power production of downwind generators in simulations (Fleming et al., 2016; Gebraad et al., 2017; Fleming et al., 2018; Archer and Vasel-Be-Hagh, 2019) and wind tunnel experiments (Adaramola and Krogstad, 2011; Mühle et al., 2018; Bastankhah and Porté-Agel, 2019). Further, the potential for wake steering to increase wind farm power production in wind conditions with wake losses has been observed in full-scale field campaigns with two (Fleming et al., 2017, 2019) and six wind turbines (Howland et al., 2019).

While wake steering has been shown to be a beneficial global wind farm control strategy compared to greedy operation, the selection of the optimal yaw misalignment strategy for each wind turbine at a farm is challenging. The optimal yaw misalignment angles depend on the wake interactions between wind turbines (Gebraad et al., 2017). These wake interactions are dependent on wind speed, wind direction, atmospheric stability, turbulence intensity, local terrain, and other flow features (see e.g. Hansen et al., 2012). Most wake steering control strategies have relied on static engineering wake models such as the FLORIS model (Gebraad et al., 2016a, b; Fleming et al., 2016) or a lifting line model (Shapiro et al., 2018; Howland et al., 2019) to select the optimal yaw misalignment strategy based on a steady, time-averaged assumption of the wind farm flow. Extremum seeking wake steering control has also been tested by Campagnolo et al. (2016b) using a gradient-based controller. However, these static model approaches may have challenges in establishing the optimal yaw misalignment strategy as a function of time in a transient flow environment such as the stable atmospheric boundary layer (ABL) or the full diurnal cycle (see e.g. Wyngaard, 2010).

Recent work has focused on the selection of the optimal yaw misalignment angles as a function of time for transient flow applications. The main challenge in this flow environment is in accurately predicting the power production given greedy baseline control, considering ABL and controller state uncertainty in a utility-scale wind farm. The combined two-dimensional computational fluid dynamics and adjoint-based optimization model WFSim has been utilized (see e.g. Boersma et al., 2016, 2018; Vali et al., 2019) for control applications. Subsequent studies have used the Ensemble Kalman filter (EnKF) to perform model state estimation as a function of time (Doekemeijer et al., 2017). The EnKF filter has been successfully used for low-order model state estimation for the purpose of receding horizon frequency regulation control (Shapiro et al., 2017) and reference power signal tracking applications (Shapiro et al., 2019). Doekemeijer et al. (2018) found that the EnKF has comparable state estimation performance given either nacelle-mounted LiDAR data or Supervisory Control and Data Acquisition (SCADA) power production data alone. Since very few utility-scale wind turbines have nacelle-mounted LiDAR systems, the successful performance of the EnKF based on SCADA data alone highlights the potential for online model calibration without additional hardware installation. The present computational complexity of the EnKF-WFSim system may limit the applicabil-



ity to real-time control systems, although recent efforts have focused on the reduction of the simulation time of this method. As such, real-time closed-loop wind farm controllers with online state estimation require reliable analytic wake models such as the FLORIS or lifting line model.

Static wake model based dynamic control studies have utilized a quasi-static wake steering approach wherein the optimal yaw misalignment angles are computed and stored as a function of wind speed and direction based on static wake models

with pre-defined model parameters (Fleming et al., 2019). However, the pre-defined model parameters were calibrated for the FLORIS wake model based on idealized LES and their applicability to a new utility-scale field implementation are unknown *a priori*. Further, there is additional uncertainty associated with the freestream velocity and turbulence intensity measurements in a wind farm environment where the typical sensors are limited to nacelle-mounted anemometers placed directly behind the rotating rotor. The dynamic influence of yaw misalignment on these sensors is unknown (Howland et al., 2019). Recently,

Raach et al. (2019) used the FLORIS wake model to design a closed-loop wake steering controller which relies on a downwind facing nacelle-mounted LiDAR system which was able to increase power production in an example nine wind turbine LES case. In order to focus on a low-order methodology which does not require additional hardware installations, we develop closed-loop wake model based wake steering control for the application of data-driven wind farm power maximization based on SCADA power production data. The algorithm was designed for real-time control of utility-scale wind turbines without the requirement

of additional hardware or sensor measurement systems and utilizes the gradient-based optimal yaw algorithm developed by Howland et al. (2019). The dynamic wake steering controller implemented in this study does not require historical data to be sorted into pre-selected wind speed and direction bins in order to make optimal yaw misalignment decisions. This is beneficial since the sorting of SCADA data represents a major uncertainty associated with wake steering control (Fleming et al., 2019; Howland et al., 2019).

Analytic wake models require a number of simplifications of the flow physics and wind turbine operation in order to predict wind farm power production in a computationally efficient fashion (see e.g. review by Stevens and Meneveau, 2017). The selected model-based optimal yaw misalignment angles will depend on the wake deflection model form and parameters, and the model for power production degradation as a function of the yaw misalignment angle. Further, in a low-order, model-driven power optimization application, the selected yaw misalignment angles will depend on the wind farm layout, wind direction and

speed, and stability state of the ABL. The goal of the present study is to analyze the sensitivity of wind farm power production increases through wake steering as a function of the design of the control system, model for power loss as a function of yaw misalignment, and wind farm layout. This work represents Part 1 of the results and targets a canonical planetary boundary layer with conventionally neutral stratification. Part 2 will focus on a sensitivity analysis of wake steering control as a function of the temporally varying stratification and surface heat flux. Section 2 will introduce the dynamic wake steering methodology

and EnKF state estimation technique. The LES methodology is introduced in Section 3. The dynamic wake steering will be validated in a two-turbine uniform inflow LES case in Section 4. In Section 5, the sensitivity to model architecture and parameters as well as wind farm layout is tested in LES of the conventionally neutral ABL with realistic Coriolis forcing. Finally, conclusions are given in Section 6.



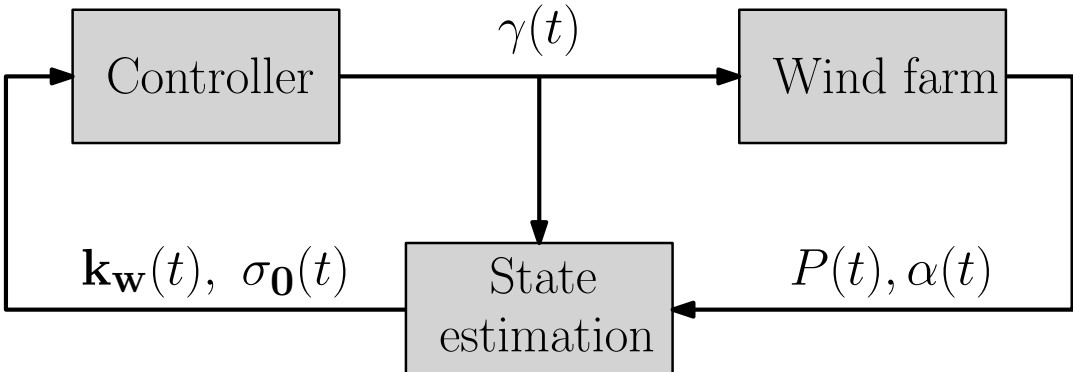

**Figure 1.** Diagram of the dynamic wake steering control system. The wake model parameters as a function of time are $k_w(t)$ and $\sigma_0(t)$ and the yaw misalignment angles are given by $\gamma(t)$. The power production and wind direction are given by $P(t)$ and $\alpha(t)$, respectively. In open-loop control, the model parameters $k_w$ and $\sigma_0$ are fixed as a function of time.

## 2 Dynamic wake steering methodology

The present methodology is focused on optimal closed-loop wake steering control as a function of time for transient flow applications. The dynamic wake steering controller is illustrated in Figure 1. The controller entails a forward-pass wake model described in Section 2.1 and a backward pass to compute analytic gradients for gradient-ascent power maximization (Section 2.3). State estimation uses the ensemble Kalman filter described in Section 2.2. The wind farm is simulated using LES (Section 3).

### 2.1 Lifting line wake model

Following the observation of counter-rotating vortex pairs shed by wind turbines operating in yaw misalignment in experiments and LES (Howland et al., 2016), Shapiro et al. (2018) developed a wake model for wind turbines in yaw based on Prandtl's lifting line theory. The wake model derived by Shapiro et al. (2018) was reformulated by Howland et al. (2019) to improve computational efficiency and to extract analytic gradients for the purpose of gradient-based optimization. Readers are directed
to Shapiro et al. (2018) for the derivation of the initial wake model and to Howland et al. (2019) for the analytic formulation which eliminates the need for domain discretization. In the two dimensional static wake model, the effective velocity at a downwind wind turbine $j$ is given as

$$u_{e,j}(x) = u_\infty - \sum_i^{N_f} \frac{\sqrt{2\pi}\delta u_i(x)d_{w,i}(x)D}{16\sigma_{0,i}} \left[ \text{erf}\left( \frac{y_T + D/2 - y_{c,i}(x)}{\sqrt{2}\sigma_{0,i}d_{w,i}(x)} \right) - \text{erf}\left( \frac{y_T - D/2 - y_{c,i}(x)}{\sqrt{2}\sigma_{0,i}d_{w,i}(x)} \right) \right], \quad (1)$$

where $u_\infty$ is the incoming freestream velocity and $\delta u_i$ and $d_{w,i}$ are the velocity deficit and the wake diameter as functions of
$x$ associated with the upwind turbine $i$, respectively. The wind turbine rotor diameter is given by $D$. The downwind turbine lateral centroid is $y_T$ and the lateral wake centroid is $y_{c,i}$. The wake model parameters are $k_w$, the wake spreading coefficient





and $\sigma_0$, the proportionality constant for the presumed Gaussian wake. The velocity deficit trailing a single wind turbine is

$$\delta u_i(x) = \frac{\delta u_{0,i}}{d_{w,i}^2(x)} \frac{1}{2} \left[ 1 + \mathrm{erf}\left( \frac{x}{\sqrt{2}D/2} \right) \right],$$ (2)

with $\delta u_{0,i} = 2a_i u_\infty$ and axial induction factor $a_i = 1/2 \left( 1 - \sqrt{1 - C_{T,i}\cos^2(\gamma_i)} \right)$. The thrust coefficient is given by $C_T$ and the yaw misalignment angle is given by $\gamma$. Positive and negative yaw misalignment are defined as counter-clockwise and clockwise rotations, respectively, when viewed from above. The inflow wind angle is given by $\alpha$, where $0°$ is north and proceeds clockwise to $360°$ at north again. The thrust force in the streamwise direction is assumed to follow actuator disk theory as $\cos^2(\gamma)$. The wake diameter as a function of the streamwise location $x$ is $d_{w,i}(x) = 1 + k_{w,i}\log\left(1 + \exp[2(x/D - 1)]\right)$. Linear superposition of the individual wakes is assumed in Eq. 1 (Lissaman, 1979).

The wake centerline $y_{c,i}$ is given by

$$y_{c,i} = \int_{x_{0,i}}^{x} \frac{-\delta v_i(x')}{u_\infty} dx',$$ (3)

where the spanwise velocity $\delta v$ is given similar to Eq. 2 with the initial disturbance given analytically as (Shapiro et al., 2018)

$$\delta v_{0,i} = \frac{1}{4} C_{T,i} u_\infty \cos^2(\gamma_i)\sin(\gamma_i).$$ (4)

The wind turbine model power is computed as

$$\hat{P}_i = \frac{1}{2}\rho A_i C_p u_{e,i}^3,$$ (5)

where $A$ is the wind turbine rotor area and $\rho$ is the density of the surrounding air. The model for the coefficient of power $C_p$ as a function of the yaw misalignment remains an open question. Often, the power loss as a function of the yaw misalignment is assumed to follow $P_{\mathrm{yaw}} \sim P\cos^{P_p}(\gamma)$, where $P_p$ is a known parameter. Following actuator disk theory (Burton et al., 2011), $P_p = 3$. However, simulations have shown for the NREL 5 MW turbine that $P_p = 1.88$ (Gebraad et al., 2016a). Recent work has shown that $P_p$ differs for freestream and waked turbines (Liew et al., 2019). The value of $P_p$ that results in a satisfactory agreement with experimental data depends on the wind turbine model, ABL shear and veer, and atmospheric stability. In the present study, we will consider $P_p$ an uncertain parameter and perform sensitivity analysis on it. The uncertainties of the wake model parameters $k_w$ and $\sigma_0$ are considered by the state estimation in Section 2.2. The coefficient of power is modeled as

$$C_{P,i} = 4\eta a_{p,i}(1 - a_{p,i})^2 \cos^{P_p}(\gamma_i),$$ (6)

with $a_{p,i} = \frac{1}{2}\left(1 - \sqrt{1 - C_{T,i}}\right)$ The parameter $\eta$ is tuned to match the manufacturer provided yaw-aligned $C_P$ look-up table (Gebraad et al., 2016a). The applicability of this model is limited to Region II of the wind turbine power curve which is typically between $4$ and $15$ m/s.

## 2.2 Ensemble Kalman filter state estimation

Engineering wake models rely on parameters which represent physical phenomena, such as the wake spreading rate $k_w$. Niayifar and Porté-Agel (2016) proposed the dependence of $k_w$ on the turbulence intensity which may be measured at a wind




farm by nacelle anemometers, and this model has been used in subsequent FLORIS applications (see e.g. Fleming et al., 2018). Shapiro et al. (2019) proposed the use of canonical turbulent wake mixing and a prescribed mixing length model to estimate $k_w$. Schreiber et al. (2019) utilized a data-driven approach where error terms are added to the engineering model and SCADA is used for data assimilation to correct the wake model inaccuracies. Gradient optimization-based SCADA data assimilation was used by Howland et al. (2019) to select the model parameters which minimize the model error in producing the site-specific wind farm greedy baseline power production. Howland and Dabiri (2019) subsequently used gradient descent coupled with a genetic algorithm for data assimilation.

Here, we will employ the EnKF (Evensen, 2003) state estimate technique along with the wake model described in Section 2.1. The EnKF filter was found to be computationally less expensive than the gradient-based data assimilation used by Howland et al. (2019). The EnKF filter is computationally superior to other Kalman filter methods (extended, unscented, etc.) since there are typically fewer ensemble states than dimensions but this may lead to spurious correlations in the state representation (see e.g. discussion by Doekemeijer et al., 2018; Mandel, 2009). The states and dimensions here represent the wake model instantiations and parameters, respectively. In our state estimation case, the dimension space scales linearly with the number of turbines $N_T$, rather than with the $N_T^2$ or $N_T^3$ in a model with a domain discretization (see discussion by Howland et al., 2019). Therefore the number of ensembles, which is a hyperparameter selected by the EnKF user, and dimensions will be of the same order of magnitude. The SCADA power production of each wind turbine is a function of time, denoted $P_k$, where $k$ is the time step index. The goal is to estimate the wake model parameters given SCADA power production data measurements, $P_k \in \mathrm{IR}^{N_t}$. This approach follows previous uses of the EnKF for wake model state estimation (Shapiro et al., 2017; Doekemeijer et al., 2017) but the algorithm is reviewed here. The EnKF is particularly well-suited for discretized partial differential equation systems, often in geophysical applications, and is computationally efficient for the present application as well. There are two wake model parameters for each upwind turbine and no parameters for the last turbine downwind. The model parameters with $N_t$ wind turbines at the $k^{th}$ time step are given by

$$\psi_k = [k_{w,1},...,k_{w,N_t},\sigma_{0,1},...,\sigma_{0,N_t}]. \tag{7}$$

The modeling and measurement errors are represented by $\chi = [\chi_{k_w}^T, \chi_{\sigma_0}^T]^T \in \mathrm{IR}^{2N_t}$ and $\varepsilon \in \mathrm{IR}^{N_t}$, respectively. The modeling errors $\chi_{k_w}$ and $\chi_{\sigma_0}$ are zero mean and have prescribed variances of $\sigma_{kw}^2 = 0.0009$ and $\sigma_{\sigma_0}^2 = 0.0009$. The Gaussian random measurement noise $\varepsilon$ has zero mean and a prescribed standard deviation of $\sigma_\varepsilon = 0.03 \cdot P_1$. The hyperparameter variances were selected based on tuning experiments (see Appendix A). In order to estimate the state model parameters, the EnKF filter uses an ensemble of wake model evaluations. The ensemble is given by

$$\Psi = [\psi^{(1)},...,\psi^{(N_e)}] \in \mathrm{IR}^{2N_t \times N_e}. \tag{8}$$

The power predictions are given by the matrix

$$\hat{\Pi} = [\hat{\pi}^{(1)},...,\hat{\pi}^{(N_e)}] \in \mathrm{IR}^{N_t \times N_e}, \tag{9}$$

where $N_e$ is the number of ensembles.



The statistical noise of the power production measurements is given by $\varepsilon$. The Gaussian random noise is added to the SCADA measurements for each ensemble

$$\xi^{(i)} = P_{\text{data}} + \varepsilon^{(i)}. \tag{10}$$

The perturbed power production ensemble matrix is

$$\Xi = [\xi^{(1)}, ..., \xi^{(N_e)}] \tag{11}$$

with the perturbation matrix prescribed by

$$\Sigma = [\varepsilon^{(1)}, ..., \varepsilon^{(N_e)}] \tag{12}$$

The mean of the ensemble states and modeled power production is given by

$$\overline{\Psi} = \Psi \mathbf{1}_{N_e} \tag{13}$$

$$\overline{\hat{\Pi}} = \hat{\Pi} \mathbf{1}_{N_e} \tag{14}$$

where $\mathbf{1}_{N_e} \in \mathrm{IR}^{N_e \times N_e}$ is a full matrix where all entries are $1/N_e$. The perturbation matrices are

$$\Psi' = \Psi - \overline{\Psi} \tag{15}$$

$$\hat{\Pi}' = \hat{\Pi} - \overline{\hat{\Pi}}. \tag{16}$$

The first step in the EnKF process is an intermediate forecast step

$$\Psi_{k+} = [\psi_k^{(1)} + B\chi_k^{(1)}, ..., \psi_k^{(N_e)} + B\chi_k^{(N_e)}] \tag{17}$$

$$\hat{\Pi}_{k+} = [h(\pi_k^{(1)} + B\chi_k^{(1)}), ..., h(\pi_k^{(N_e)} + B\chi_k^{(N_e)})]. \tag{18}$$

where matrix $B \in \mathrm{IR}^{2N_t \times 2N_t}$ is the identity matrix and $h$ represents the nonlinear wake model described in Section 2.1.

The measurement analysis step is given by

$$\Psi_{k+1} = \Psi_{k+} + \Psi'_{k+} \hat{\Pi}'^{T}_{k+} (\hat{\Pi}'_{k+} \hat{\Pi}'^{T}_{k+} + \Sigma_{k+1} \Sigma^{T}_{k+1})^{-1} \cdot (\Xi_{k+1} - \hat{\Pi}_{k+}). \tag{19}$$

The final values of $k_w$ and $\sigma_0$ for the $k+1$ time step are given as the columns of $\overline{\Psi}_{k+1}$. The EnKF state estimation then assumes that the parameters $k_{w,k+1}$ and $\sigma_{0,k+1}$ will be valid over the succeeding finite time from step $k+1$ until step $k+2$.

The EnKF is a Kalman filter method which uses the Monte-Carlo sampling of model parameters according to a prescribed Gaussian function to represent the covariance matrix of the probability density function (PDF) of the state vector $\Psi$. The likelihood of the data is represented using observations $\Xi$ and prescribed perturbations $\Sigma$. Using the prior PDF of the state $(k)$ and data likelihood, the posterior state $(k+1)$ is estimated using Bayes's rule (Eq. 19).



### 2.3  Optimal yaw misalignment optimization

The optimal yaw misalignment angles depend on the wind speed, direction, turbulence intensity, and other key ABL conditions.

Within a given condition bin, the number of potential yaw misalignment angle combinations grows exponentially with the number of wind turbines. As such, brute force optimization methods are not sufficient for the selection of the optimal yaw misalignment strategy. Previous studies have considered genetic algorithms (Gebraad et al., 2016a), discrete gradient-based optimization (Gebraad et al., 2017), and analytic gradient-based optimization (Howland et al., 2019). Using gradient-based ADAM optimization (Kingma and Ba, 2014), the gradient update is given by

$$\gamma^{t+1} = \gamma^t - \alpha \frac{m_t}{\sqrt{v_t}}, \tag{20}$$

where $m^t = \beta_1 m^{t-1} + (1-\beta_1)\frac{\partial \sum \hat{P}}{\partial \gamma}$ and $v^t = \beta_2 v^{t-1} + (1-\beta_2)(\frac{\partial \sum \hat{P}}{\partial \gamma})^2$. The hyperparameters are set to the commonly used values of $\beta_1 = 0.9$ and $\beta_2 = 0.999$, respectively (Kingma and Ba, 2014). The analytic gradients computed by Howland et al. (2019) are used for the gradient-based wind farm power optimization.

### 2.4  Approximate advection timescale

Upon the yaw misalignment of an upwind turbine, there is a time lag associated with the advection time scale of the flow for the control decision to influence a downwind turbine. While the advection time depends on the length scale of the turbulent eddy (Del Álamo and Jiménez, 2009; Yang and Howland, 2018; Howland and Yang, 2018), the mean flow advection approximately follows the mean wind speed in wind farms (Taylor, 1938; Lukassen et al., 2018). The number of simulation time steps associated with the approximate advection time between the first and last turbines is computed as

$$T_{\mathrm{a}} = \frac{\Delta s_x}{\overline{u}_{\mathrm{hub}} \Delta t}, \tag{21}$$

where $\Delta s_x$ is the distance between the first and last turbine in the streamwise direction and $\overline{u}_{\mathrm{hub}}$ is the mean streamwise velocity at the wind turbine model hub height at the leading turbine in the farm. The simulation time step is fixed and is $\Delta t$, which corresponds to a CFL of less than 1 persistently during runtime. In the computation of wind farm statistics for the utilization of static wake models, the advection time scale is accounted for by initializing the time averaging two advection

time scales $2T_a$ after the yaw misalignments for the wind turbine array have been updated. To account for errors associated with the simple advection model, the time lag is taken as double the advection time scale, $2T_a$. The sensitivity of the wind farm power production and model-predicted optimal yaw misalignment angles as a result of the advection time lag are considered in Section 5.

### 2.5  Yaw misalignment temporal update frequency

While static wake models are able to capture time averaged wind farm dynamics in stationary flows (see e.g. Stevens and Meneveau, 2017), instantaneous wind speed and direction are constantly changing and challenging to predict. As the stability of the atmosphere transitions during the diurnal cycle, the mean wind conditions as well as turbulence intensity will change





with a significant impact on the wake loss magnitude (Hansen et al., 2012). The wake steering strategy must be dynamic to adapt to the instantaneous wind conditions but also requires some time lag according to the advection time scale of the wind

farm. The selection of the optimal yaw misalignment angle update frequency will impact the power production of the wind farm. Kanev (2020) found that when utilizing a dynamic wake steering controller in transient flow environments the energy production may decrease as a result of wind direction fluctuations as a function of time. The energy loss was due to the dynamic wake steering controller attempting to follow the wind direction constantly as a function of time, leading to increased yaw duty, and a final yaw update time of 2 minutes was selected.

In a full-scale wake steering field experiment, Fleming et al. (2019) found that wind direction and speed data low pass filtering resulted in an unintended time lag between the observed conditions and the associated optimal yaw misalignment angles. Finally, Raach et al. (2019) utilized a feedforward-feedback framework to adjust the open-loop predicted optimal yaw misalignment angles continuously based on LiDAR measurements of the wake centroid location, instead of SCADA power measurements, with an update time scale on the order of seconds. In the present study, the yaw misalignment update frequency

is selected according to the dynamics of the problem studied. Comments on the update frequency for the conventionally neutral ABL LES cases are made in Section 5.

## 3  Large eddy simulation setup

Large eddy simulations are performed using the open-source psuedo-spectral code *PadéOps*[1]. The solver uses $6^{th}$ order compact finite differencing in the vertical direction (Nagarajan et al., 2003) and Fourier collocation in the horizontal directions.

Temporal integration uses a fourth order strong stability preserving Runge-Kutta variant (Gottlieb et al., 2011). The LES code has previously been utilized for high Reynolds number ABL flows (Howland et al., 2020b; Ghaisas et al., 2020) and is described in detail by Ghate and Lele (2017). The ABL is modeled as an incompressible, high Reynolds number limit ($Re \to \infty$) flow with the nondimensional momentum equations given by

$$\frac{\partial u_i}{\partial t} + u_j \frac{\partial u_i}{\partial x_j} = -\frac{\partial p}{\partial x_i} - \frac{\partial \tau_{ij}}{\partial x_j} + f_i + \frac{\delta_{i3}}{Fr^2}(\theta - \theta_0) - \frac{2}{Ro}\varepsilon_{ijk}\Omega_j u_k - \frac{\partial P^G}{\partial x_i}, \tag{22}$$

where $u_i$ is the velocity in the $x_i$ direction, $p$ is the nondimensional pressure, and $P^G$ is the nondimensional geostrophic pressure. The subfilter scale stress tensor is given by $\tau_{ij}$ and the sigma model is employed (Nicoud et al., 2011). The turbulent Prandtl number used in the subfilter scale model is $Pr = 0.4$ (Ghate and Lele, 2017). Surface stress and heat flux is computed using a local wall model based on Monin-Obukhov similarity theory with appropriate treatment based on the state of stratification (Basu et al., 2008). The wind turbine forcing is represented by $f_i$ and an actuator disk model is used (Calaf

et al., 2010). While the actuator disk model is lower fidelity than the actuator line methods, it captures the far wake accurately for both aligned (Martínez-Tossas et al., 2015) and yaw misalignment wind turbines (Lin and Porté-Agel, 2019). Since the goal of the present study is controller synthesis and sensitivity experiments, more computationally expensive actuator line simulations are left for future work given the large volume of simulations that are run. Earth's rotational vector is given by

---

[1]https://github.com/FPAL-Stanford-University/PadeOps





$\Omega = [0, \cos(\phi), \sin(\phi)]$, where $\phi$ is the latitude. The traditional approximation, which neglects the horizontal component of

Earth's rotation (Leibovich and Lele, 1985), is not enforced. Therefore, Earth's full rotational vector is included resulting in wind farm dynamics which are sensitive to the direction of the geostrophic wind (Howland et al., 2020a). For simplicity all simulations are performed with west to east geostrophic wind. The Coriolis terms are parameterized by the Rossby number $Ro = G/\omega L$, where $G$ is the geostrophic wind speed magnitude, $\omega$ is Earth's angular velocity, and $L$ is the relevant length scale of the problem. All wind speeds used in this study will be normalized by the geostrophic wind speed magnitude. The

nondimensional potential temperature is given by $\theta$. The buoyancy term is parameterized by the Froude number $Fr = G/\sqrt{gL}$, where $g$ is the gravitational acceleration. The equation for the transport of the filtered nondimensional potential temperature is given by

$$\frac{\partial \theta}{\partial t} + u_j \frac{\partial \theta}{\partial x_j} = -\frac{\partial q_j^{SGS}}{\partial x_j}, \tag{23}$$

where $q_j^{SGS}$ is the subgrid scale (SGS) heat flux.

The wind is forced by prescribing the geostrophic approximation where the geostrophic pressure gradient drives the mean flow (Hoskins, 1975). The geostrophic pressure balance in the stable free atmosphere is given by

$$\frac{\partial P^G}{\partial x_i} = -\frac{2}{Ro} \varepsilon_{ijk} \Omega_j G_k, \tag{24}$$

with $G_k$ representing the geostrophic velocity vector.

The simulations utilize a fringe region to force the inflow to a desired profile (Nordström et al., 1999). In the uniform inflow

cases, the fringe region forces the flow to a uniform profile. In the conventionally neutral ABL cases, the concurrent precursor method is applied wherein a separate LES of the ABL is run without wind turbine models and the fringe region is used to force the primary simulation outflow to match the concurrent precursor simulation outflow (see e.g. Munters et al., 2016; Howland et al., 2020b).

For the uniform inflow and conventionally neutral cases, there is an initial startup transience following the domain initializa-

tion. The uniform inflow domain is initialized with $u = 1$ in the streamwise direction. Detailed comments on the initialization for the conventionally neutral case are given by Howland et al. (2020a). The simulation cases are run until statistical stationarity and quasi-stationarity is reached for the uniform and conventionally neutral cases, respectively. The conventionally neutral case is statistically quasi-stationary due to inertial oscillations (see Allaerts and Meyers, 2015, for a detailed discussion on the conventionally neutral ABL quasi-stationarity). Upon convergence, the wake steering control strategy is initiated.

The control is initialized with greedy baseline yaw alignment which is fixed for $n_T$ time steps. After $n_T$ simulation steps, with the time averaged power production for each wind turbine measured over the previous $n_T - 2T_a$ time steps, the EnKF state estimation and optimal yaw calculations are performed (Figure 1). The yaw angles are then implemented and held fixed for $n_T$ time steps and the cycle repeats. The wind speed, wind direction, and power production are averaged in time over the window. The state estimation and yaw misalignment update steps are performed concurrently with a period of $n_T$ simulation

steps. In general, these two processes can be decoupled, although this was not investigated in the present study.



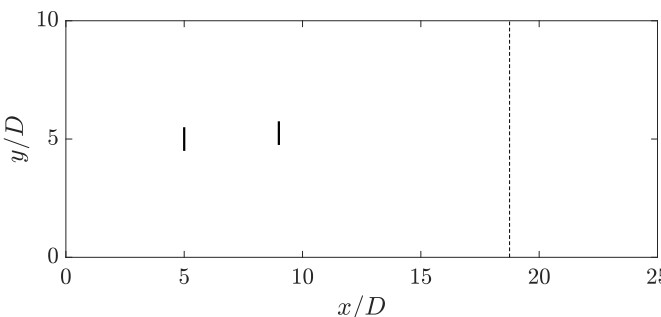

**Figure 2.** Uniform inflow LES simulation actuator disk model layout. The fringe region is represented by the dashed black line and the wind turbines are shown as solid black lines.

In order to compare the power production of the yaw misalignment control strategy with the baseline greedy control, a separate LES case is run for each experiment with yaw aligned control. The two simulations are initialized from identical domain realizations and the computational timestep $\Delta t$ is fixed between the two cases. Therefore, without the influence of variable turbine operation, the flow within and around the turbine array is identical to machine precision between the two yaw

aligned and yaw misaligned cases. Since this study will consider the conventionally neutral ABL which contains turbulence and inertial oscillations, this separate simulation must be used instead of a comparison with the power production of the first yaw control update step (see Appendix C).

## 4   Dynamic wake steering uniform inflow LES

In this section, the dynamic closed-loop wake steering controller described by Figure 1 will be used in LES of two turbines

operating in uniform inflow. The domain has lengths of $25D$, $10D$, and $10D$ in the $x$, $y$, and $z$ directions, respectively, and the number of grid points are 128, 64, and 64. Two actuator disk model wind turbines are simulated in uniform inflow with slip walls on all sides and a fringe region at the domain exit to force the inflow to a uniform profile. The fringe is used in the last $25\%$ of the computational $x$ domain. The turbines are located $4D$ apart in the streamwise direction and are misaligned by $0.25D$ in the spanwise direction as shown in Figure 2. Due to the spanwise misalignment, the preferential yaw misalignment

direction for the upwind turbine is positive (counter-clockwise rotation viewed from above).

The flow is stationary after the initial startup transient has decayed and therefore the optimal yaw misalignment angles for the two wind turbines are not a function of time. The flow is initialized as described in Section 3. Upon statistical stationarity, the closed-loop wake steering controller is initialized and the flow is run for $n_T = 10000$ LES time steps to ensure sufficient averaging. The time averaging is initialized following the advection time of the wind farm (see Section 2.4) and therefore there

are $n_T - 2T_a$ timesteps within each time averaging window.

The sum of power production for the two turbine pair as a function of the control update steps is shown in Figure 3(a). The power production is normalized by the greedy control simulation. The power production for the first yaw controller update time





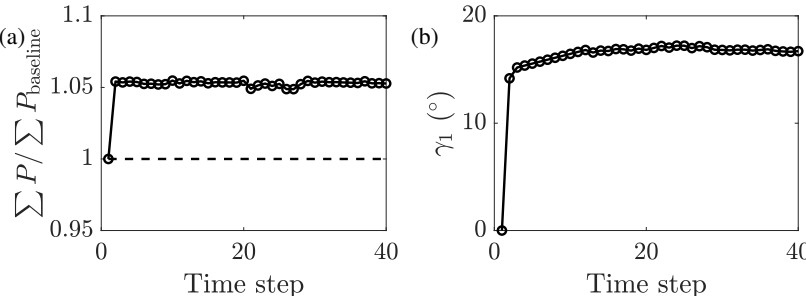

**Figure 3.** Two wind turbine model uniform inflow LES (a) power production. (b) The yaw misalignment angle $\gamma$ for the upwind turbine. The downwind turbine remains yaw aligned during the simulation.

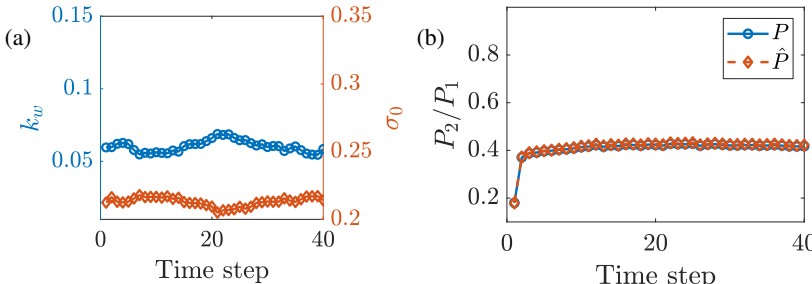

**Figure 4.** Two wind turbine model uniform inflow LES (a) $k_w$ and $\sigma_0$ for the upwind turbine and (b) $\hat{P}$ and $P$ for the downwind turbine.

step is equal to 1 since yaw misalignment has not been implemented and the model is gathering power production data to be used for the first EnKF data assimilation pass. The power production increases in the second time step when yaw misalignment

in incorporated for the upwind turbine (Figure 3(b)). The controller correctly commands the upwind turbine to positive yaw misalignment. While the flow is stationary, the upwind turbine yaw misalignment angle changes marginally after the second time step. These changes can be attributed to modifications to the wake model parameters as a function of time as estimated by the EnKF (Figure 4(a)). The estimated model parameters vary in time in this stationary flow due to standard error of the mean with limited samples within a given time window of length $T$, due to the influence of the yaw misalignment of the dynamics

on the wake, and due to the limited number of ensembles $N_e$ in the EnKF.

     The wake model parameters have a functional dependence on the yaw misalignment of the turbines within the wind farm. The wake of a yaw misaligned turbine is narrower than the same turbine when yaw aligned (Archer and Vasel-Be-Hagh, 2019). The wake spreading rate $k_w$ dictates the wake recovery rate. Yaw misalignment also reduces the axial induction factor of the wind turbine and therefore affects the wake recovery. Further, since wind turbines in yaw misalignment generate large-

scale counter rotating vortices (Howland et al., 2016), the wake recovery rate will likely be enhanced in yaw misalignment (Fleming et al., 2018). As a result of these vortices, the wake will have top-down asymmetry and this will influence $\sigma_0$. While the present model neglects the vertical dimension, the development of a controls-oriented model which incorporates the curled





wake asymmetry is ongoing (Martínez-Tossas et al., 2019). Future work should characterize the influence of yaw misalignment on the wake spreading rate.

The state-estimated power production for the downwind turbine is compared to the LES power production in Figure 4(b). The downwind turbine's power production in the greedy control strategy is low (approximately $0.2P_1$) due to the freestream inflow condition and close streamwise direction spacing. The EnKF results in an accurate power production estimation using the lifting line model. While the wake model parameters are changing as a function of time (Figure 4(a)), the power production estimate for the downwind turbine is not significantly affected. The wake model parameters $k_w$ and $\sigma_0$ are anti-correlated as a function

of time. Within the two-parameter lifting line model, increasing $k_w$ or $\sigma_0$ reduces the wake effect for the downwind turbine. With the LES power production of the downwind turbine and the yaw misalignment of the upwind turbine approximately fixed, the state estimation increases one parameter and reduces the other parameter to remain consistent in the estimation of the downwind power. This indicates, similarly to the results in Appendix A, that the two parameter lifting line model may be overparameterized which may lead to overfitting. The accuracy of the EnKF state estimation and lifting line model in the

prediction of the power production in yaw misalignment will be tested in Section 5. This accuracy will implicitly measure the impact of overfitting in the model. Since there is no wake impinging on the upwind turbine, the state estimation has no impact on the power prediction of the upwind turbine since there are no wake model parameters to estimate. The accuracy of the upwind turbine model prediction will be governed by the fidelity of the cosine model and $P_p$, given by Eq. 6.

        The instantaneous streamwise velocity is visualized in Figures 5(a) and (b) for the baseline greedy yaw control and the

optimal yaw misalignment angle, respectively. As a result of the yaw misalignment, the wake is partially laterally deflected away from the downwind turbine. The yaw misalignment increases the magnitude of the streamwise velocity in the wake region. The wake region trailing the second turbine is reduced in size and intensity as a result of the yaw misalignment action of the first turbine, indicating that for larger columns of turbines the potential for power increases due to wake steering are larger.



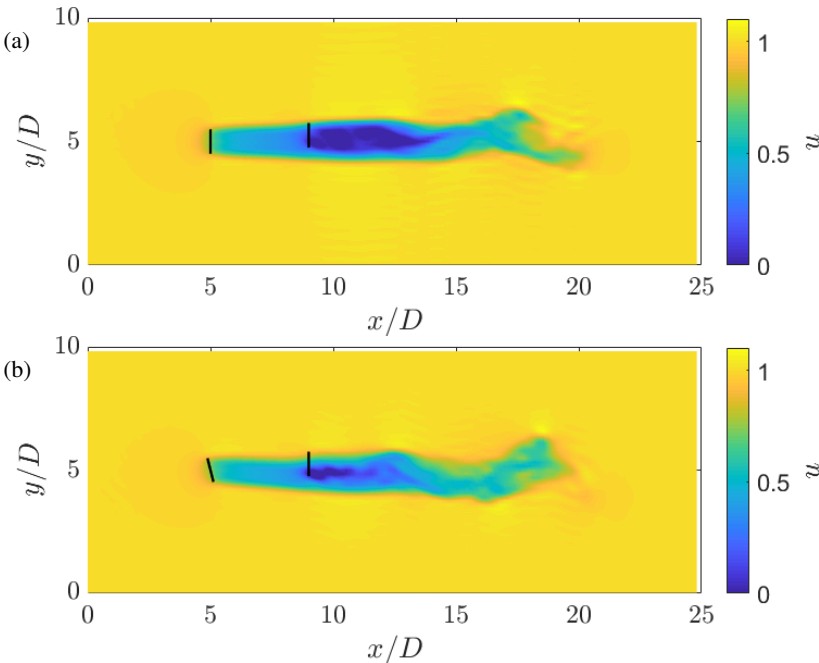

**Figure 5.** Two wind turbine model uniform inflow LES instantaneous streamwise velocity $u$ snapshot with (a) no yaw misalignment (b) after the last time step of dynamic wake steering.

## 380  5   Dynamic wake steering neutral atmospheric boundary layer LES

In this section, we will utilize the closed-loop wake steering controller that was validated in Section 4 for uniform inflow in the conventionally neutral ABL. While the conventionally neutral ABL is quasi-stationary, the optimal yaw misalignment angles will vary as a function of time due to turbulence, large-scale streamwise structures (Önder and Meyers, 2018), and inertial oscillations. A suite of LES cases is run to test the influence of the controller architecture design, state estimation design, 385  $P_p$ estimate, and the wind farm layout on the power production increases over greedy baseline operation as a result of wake steering control. Each sensitivity study represents a new LES case which is run using the concurrent precursor methodology described in Section 3.

All quasi-steady conventionally neutral ABL simulations have a yaw controller update of $n_T = 1000$ time steps which approximately equal to $\tau = 3000$ seconds or 50 minutes. The advection time scale from the first to the last wind turbine in the 390  array is approximately 9 minutes and the time lag is taken as two times the approximate advection time scale based on Taylor's Hypothesis. Therefore, each update contains approximately 30 minutes of statistical averaging, or about 600 time steps. The long time averaging window was selected since the flow is quasi-stationary and to ensure temporal averages with reduced noise. In transitioning ABL environments, the time averaging window should likely be reduced (Kanev, 2020). The greedy baseline controller yaw alignment is updated according to the same timescales based on the mean wind direction measured locally by 395  each wind turbine. The nacelle position for the yaw misaligned turbines is based on the wind direction measurement at each



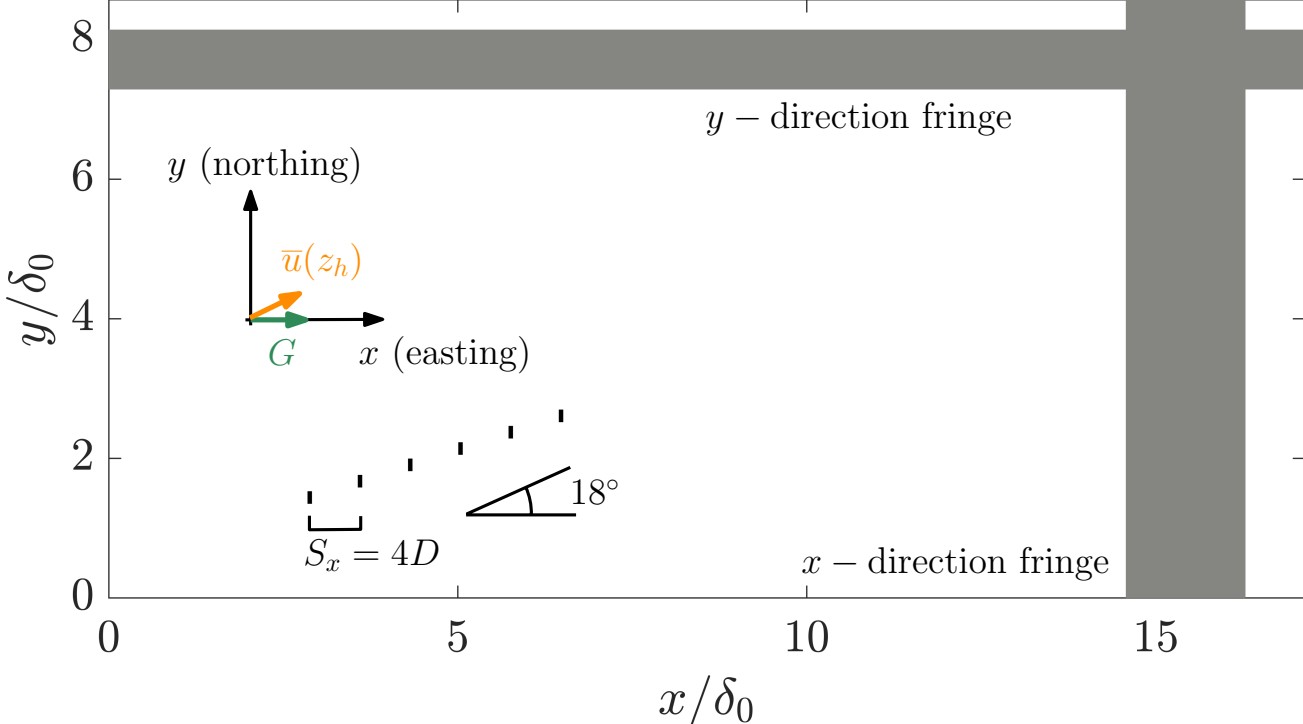

**Figure 6.** Conventionally neutral six wind turbine finite wind farm simulation setup. The geostrophic wind direction is west to east and the $x$-axis is aligned with the geostrophic wind direction. The mean wind direction at hub height, $\tan^{-1}(\overline{v}/\overline{u}) \approx 16°$ but is not known *a priori* in the simulation and varies as a function of time. The wind turbine array is offset from alignment in the $x$-direction by $18°$. The initial boundary layer height $\delta_0$ is 700 meters and does not change significantly during runtime (see Figure 7). Fringe functions are applied in the $x$ and $y$ directions to establish a finite wind farm simulation.

local turbine as well as the controller estimated optimal yaw misalignment angles, i.e. $n_\alpha = \alpha + \gamma$ where $n_\alpha$ is the nacelle position and $\alpha$ is the wind direction incident to the wind turbine.

The wind turbines have a rotor diameter of 126 meters and a hub height of 100 meters (selections based on NREL 5 MW turbine, Jonkman et al., 2009). The thrust coefficient is $C_T = 0.75$. The initial boundary layer height is 700 meters. The domain
400 size is $12 \times 6 \times 2.4$ kilometers in the $x$, $y$, and $z$ directions, respectively, with $z$ representing the wall-normal coordinate. The number of grid points is $480 \times 240 \times 192$ with a grid spacing of $25\text{m} \times 25\text{m} \times 12.5\text{m}$. The grid spacing is uniform and the mesh size is similar to previous studies (Allaerts and Meyers, 2015) and a grid convergence study was performed by (Howland et al., 2020a) for the conventionally neutral ABL. Six model wind turbines are incorporated in the domain and the layout within the computational domain is shown in Figure 6. The Rossby number based on the wind turbine diameter is 544 and the Froude
405 number is 0.14. The vertical profiles of velocity, potential temperature, and streamwise turbulence intensity for the precursor simulation for two domain snapshots are shown in Figure 7.





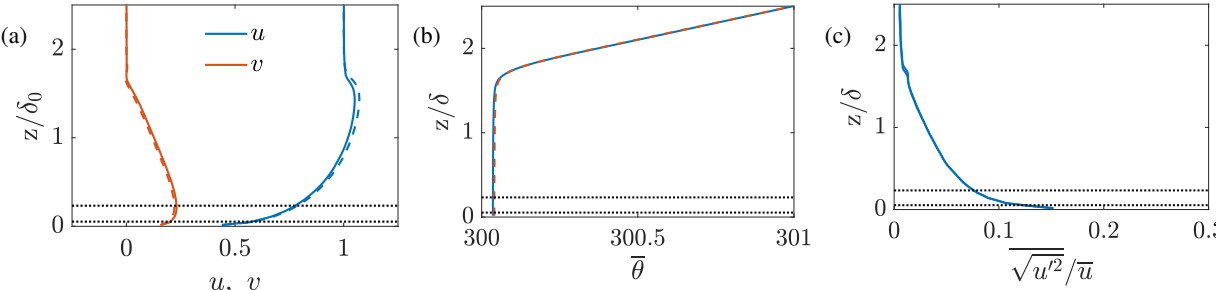

**Figure 7.** Horizontally averaged concurrent precursor conventionally neutral ABL LES (a) velocity, (b) potential temperature, and (c) turbulence intensity. Two different instantaneous domain snapshots are shown in solid (early) and dashed (late) lines. Dashed-dotted lines show the extents of the turbine rotor area.

| Case | Steering | Static yaw | Static $k_w, \sigma_0$ | Advection | Feedforward $k_w, \sigma_0$ | $P_p$ | $\frac{\sum P - \sum P_{\text{aligned}}}{\sum P_{\text{aligned}}}$ (%) | $\overline{\sum P} \pm \text{STD}(\sum P)$ |
|---|---|---|---|---|---|---|---|---|
| | | | | Wind turbine column alignment $18°$ | | | | |
| NA | - | - | - | ✓ | - | 3 | - | $3.01 \pm 0.13$ |
| NL | ✓ | ✓ | ✓ | ✓ | - | 3 | 5.4 | $3.17 \pm 0.14$ |
| ND1 | ✓ | - | - | ✓ | - | 3 | 0.2 | $3.01 \pm 0.12$ |
| ND2 | ✓ | - | - | ✓ | ✓ | 3 | 4.6 | $3.15 \pm 0.14$ |
| ND3 | ✓ | - | ✓ | ✓ | - | 3 | 5.7 | $3.18 \pm 0.13$ |
| ND4 | ✓ | - | - | ✓ | ✓ | 2 | $-3.0$ | $2.92 \pm 0.13$ |
| ND5 | ✓ | - | - | ✓ | ✓ | 4 | 5.1 | $3.16 \pm 0.16$ |
| ND6 | ✓ | - | - | - | ✓ | 3 | 4.2 | $3.13 \pm 0.12$ |
| | | | | Wind turbine column alignment $14°$ | | | | |
| NA14 | - | - | - | ✓ | - | 3 | - | $2.96 \pm 0.09$ |
| ND141 | ✓ | - | - | ✓ | ✓ | 3 | 1.1 | $2.99 \pm 0.10$ |
| ND142 | ✓ | - | ✓ | ✓ | - | 3 | 1.0 | $2.99 \pm 0.11$ |

**Table 1.** The conventionally neutral finite wind farm wake steering cases. The mean power production increase with respect to yaw aligned operation is calculated over approximately 24 hours of physical wind farm operation.

This section is organized as follows: Section 5.1 examines the sensitivity of the wind turbine array power production to the wake steering controller design. Section 5.2 tests the sensitivity to the state estimation methodology. The sensitivity of the wake steering control to $P_p$ is discussed in Section 5.3. The accuracy of the wake model power predictions is discussed in Section 5.4. Finally, Section 5.5 characterizes the influence of the wind farm alignment on the wake steering power production increase. The conventionally neutral ABL wake steering results are summarized in Table 1.



## 5.1 Comparison between dynamic and quasi-static wake steering approaches

The dynamic wake steering controller described in Figure 1 is compared to lookup table static control in this section. Since the flow is quasi-stationary, the mean wind speed and direction at hub height do not change significantly as a function of time. Therefore, during simulation, the flow remains at wind conditions which would be associated with one wind speed and direction bin in tabulated lookup table wake steering control. The lookup table control is approximated by fixing the yaw misalignment angles as a function of time after the initial optimal angles are computed during the first yaw controller update (Case NL). The dynamic yaw controller is represented by Case ND2.

The yaw misalignment angles as a function of the yaw controller updates for Cases NL and ND2 are shown in Figure 8. The lifting line model selects yaw misalignment angles which are large for the first turbine and generally decrease further into the wind farm, which is consistent with recent wind tunnel experiments (Bastankhah and Porté-Agel, 2019). Since the flow is quasi-stationary, the dynamic algorithm yaw misalignment angles do not change significantly as a function of time. There are a few yaw misalignment changes on the order of $10°$ during one yaw update. The change in yaw misalignment in a single control update is not limited explicitly in this study. The time averaged power productions as a function of the yaw controller updates for the two cases are shown in Figure 9. The qualitative trends in power production are similar between the two cases. Quantitatively, the lookup table static yaw misalignment Case NL increased the power production $5.4\%$ with respect to the baseline greedy control while the dynamic yaw Case ND2 increased the power by $4.6\%$.

The quantitative influence of wake steering is a function of the layout and ABL conditions. As the focus of the present study is assessing the sensitivity of wake steering to controller architecture, model parameters, and wind farm layout, measures of the statistical significance of the results are useful. However, the statistical significance of the results (e.g. whether Case NL significantly outperformed Case ND2) does not indicate, necessarily, that lookup table control is better than the dynamic controller used in Case ND2 for all wake steering applications but rather, that it was better for the specific ABL setup and computational time window of the experiment. In this study, we will consider a control case to be significantly superior to another if the mean array power production averaged over the control update steps is more than one standard deviation larger than the other case. The mean and standard deviations of the array power productions over the control update steps are shown in Table 1. Cases NL and ND2 have significantly higher power than Case NA but the power in Case NL is not significantly higher than in Case ND2.

Although it is not significant, there are several possible reasons for the static yaw misalignment's slightly superior performance compared to the dynamic yaw controller. The yaw selection may overfit to the previous time window and select angles which are suboptimal for the next time window (tested further in Section 5.2). The relationship between the wake model power prediction and the measured LES power production is shown for the two cases in Figure 10. The wake model overpredicts the power production in yaw misalignment more for the dynamic yaw control than the lookup table control. After the first time step, the wake model no longer has any state information for the LES power production with greedy baseline control since the previous state had yaw misalignment. When the wake model overpredicts the expected LES power, the wake model parameters are updated to a state which expects larger wake loss effects in baseline control; therefore, the yaw misalignment angles are

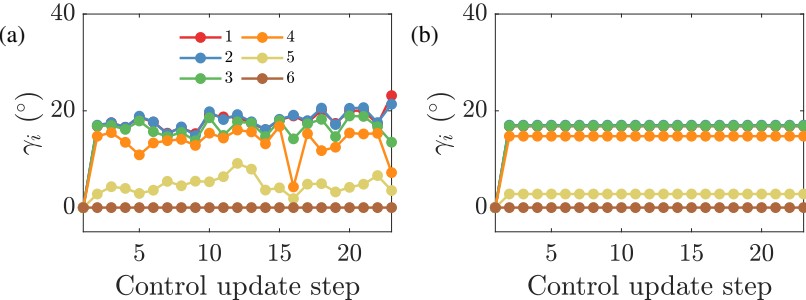

**Figure 8.** Wind farm yaw misalignment angles $\gamma_i$ for each turbine for (a) online control using the initial parameters to initialize the next state (ND2) and (b) and the lookup table control (NL).

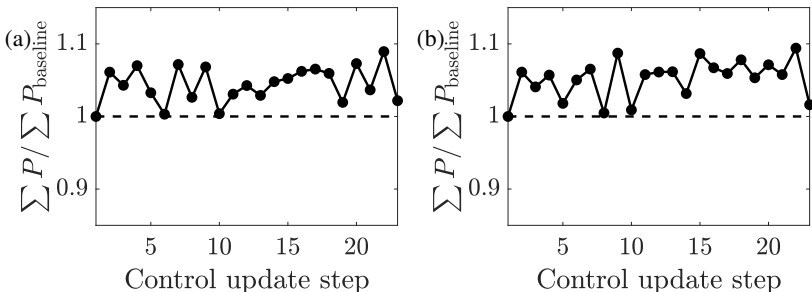

**Figure 9.** Time averaged wind farm power production as a function of the control update steps for (a) online control using the initial parameters to initialize the next state (ND2) and (b) and the lookup table control (NL). The wind farm power is normalized by the power production of the aligned wind farm case.

increased at the next time step. The yaw misalignment angles for the leading turbine oscillate around the lookup table optimal forecast which was based on calibration with power data from greedy baseline control alone (Figure 8). The dynamic yaw increased power slightly less than the static yaw misalignment case. However, eliminating the need to tabulate historical data and the complexity of implementing a lookup table-based controller could be beneficial in a practical controller setting. Further, the conventionally neutral boundary layer does not occur often in practice (Hess, 2004). Therefore, in a practical setting, the wind direction and speed at hub height will not be fixed for multiple hours as in this test problem.

In Case ND6, the power productions are time averaged over the full $n_T$ window without considering the advection time scale in the controller design. The power production increase over greedy control is $4.2\%$ in this case which is less than the $4.6\%$ increase when considering the advection time lag (Case ND2), although this difference is not significant. The dynamics of the closed-loop controller over long experimental horizons are tested in a 50 control update simulation in Appendix B.

### 5.2  Influence of the state estimation

The influence of the state estimation methodology is tested in this section. Within the conventionally neutral ABL, three experiments are run, focused on the state estimation initialization. In Case ND1, the optimal EnKF estimated parameters from





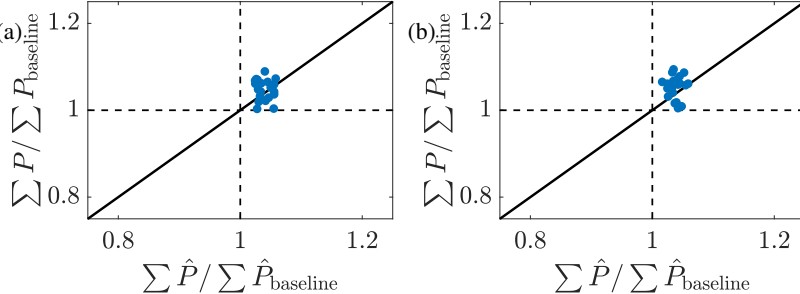

**Figure 10.** Relationship between the LES wind farm power production compared to the wake model wind farm power production prediction for (a) online control using the initial parameters to initialize the next state and (b) and the lookup table control. The wind farm power is normalized by the power production of the aligned wind farm case. The LES power production is given by $P$ and the wake model prediction is given by $\hat{P}$.

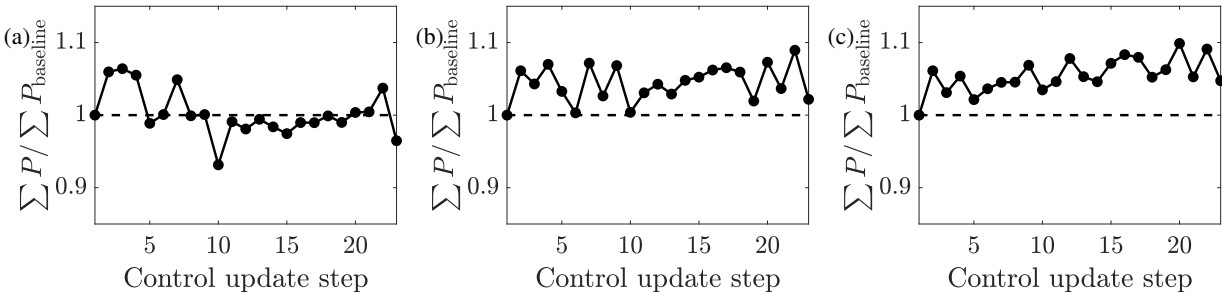

**Figure 11.** Time averaged wind farm power production as a function of the control update steps for (a) online control using the previous optimal parameters to initialize the next state (ND1), (b) online control using the initial parameters to initialize the next state (ND2), and (c) and the static state estimation control (ND3). The wind farm power is normalized by the power production of the aligned wind farm case.

the previous time step are used to initialize the state estimation of the current time step. The initial model parameters are held
fixed at $k_w = 0.1$ and $\sigma_0 = 0.25$ in Case ND2. Finally, Case ND3 fixes the model parameters after the first time step. Case ND3 differs from Case NL from Section 5.1 since the optimal yaw misalignment angles may vary as a function of time while the model parameters do not.

The power productions as a function of the yaw controller update for the three cases are shown in Figure 11. Case ND1 has significantly less power production than Cases ND2 and ND3. The time averaged power production increases with respect to the
baseline, greedy control is $0.2\%$, $4.6\%$, and $5.7\%$ for cases ND1, ND2, and ND3, respectively. The power production in Cases ND2 and ND3 are significantly higher than Case NA while Case ND1 is not. Further, Cases ND2 and ND3 are significantly better than ND1 but Case ND3 is not significantly better than ND2. In the EnKF methodology described in Section 2.2, the update step to the wake parameters is limited by the imposed parameter variance ($\sigma_{k_w}$ and $\sigma_{\sigma_0}$). Therefore, the initialization of the EnKF with fixed parameters limits the perturbation of the estimated parameters as a function of time whereas the
initialization with the previous optimal parameters allows $k_w$ and $\sigma_0$ to vary more significantly over time. The EnKF estimated $k_w$ and $\sigma_0$ for the three cases are shown in Figures 12 and 13, respectively. While the proportionality constant of the presumed





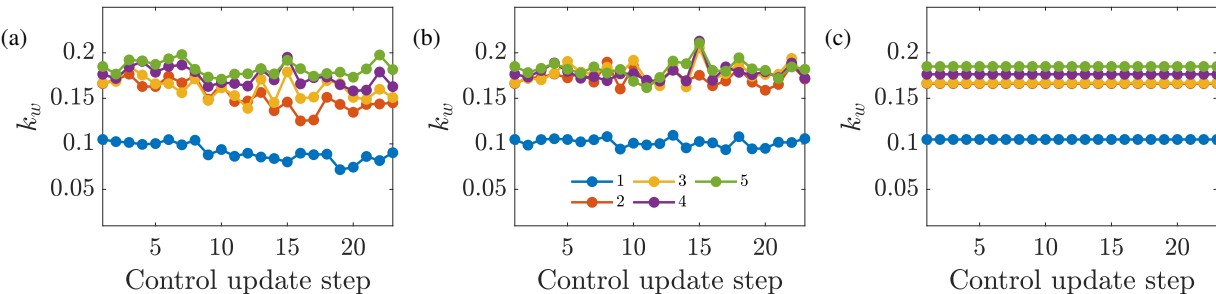

**Figure 12.** Wake spreading coefficient for each turbine in the wind farm for (a) online control using the previous optimal parameters to initialize the next state (ND1), (b) online control using the initial parameters to initialize the next state (ND2), and (c) and the static state estimation control (ND3).

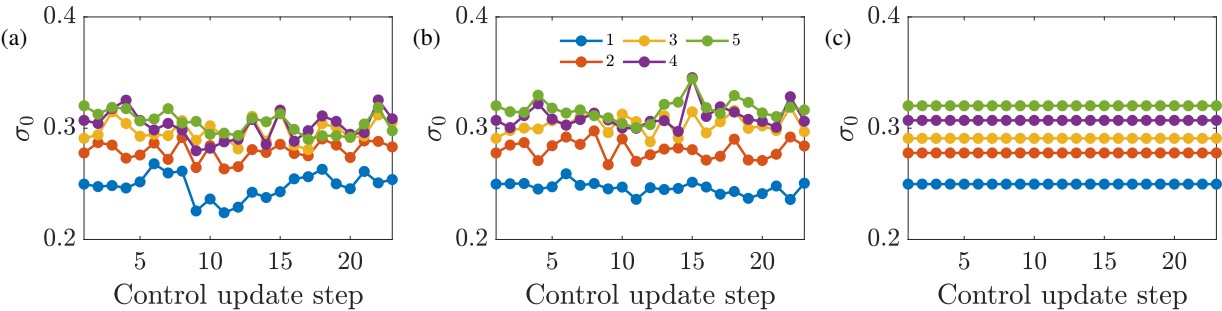

**Figure 13.** Proportionality constant for the presumed Gaussian wake for each turbine in the wind farm for (a) online control using the previous optimal parameters to initialize the next state (ND1), (b) online control using the initial parameters to initialize the next state (ND2), and (c) and the static state estimation control (ND3).

Gaussian wake does have a clear trend for Case ND1, the estimated wake spreading rate $k_w$ is clearly decreasing for all wind turbines as a function of time. For Case ND2, the estimated model parameters do not have a clear trend and remain approximately constant as a function of time. As the estimated wake spreading rate is decreased, the wake model predicts

worsening wake interactions and lower array power production given greedy baseline control. As a result, the model predicted optimal yaw misalignment angles increase as a function of time for Case ND1 as shown in Figure 14(a). While Cases ND2 and ND3 predict the optimal yaw misalignment for the most upwind turbine to be approximately $20°$, and decreasing $\gamma$ moving downwind, Case ND1 increases the yaw misalignment for the upwind turbine to as high as $30°$. While this case was not run further, it is not expected that this trend would continue unboundedly with controller instability since the power production

penalty as a function of increased yaw misalignment is significant beyond $40°$. The typical yaw rate for utility-scale horizontal axis wind turbines is around $0.5$ degrees per second. For the largest discrete yaw misalignment change in the present study of $\approx 30°$ (Figure 14(a)), the yaw misalignment change would take $\approx 75$ seconds. This time is significantly less than the advection time $T_a$, and is therefore does not impact the control system and power production results here.





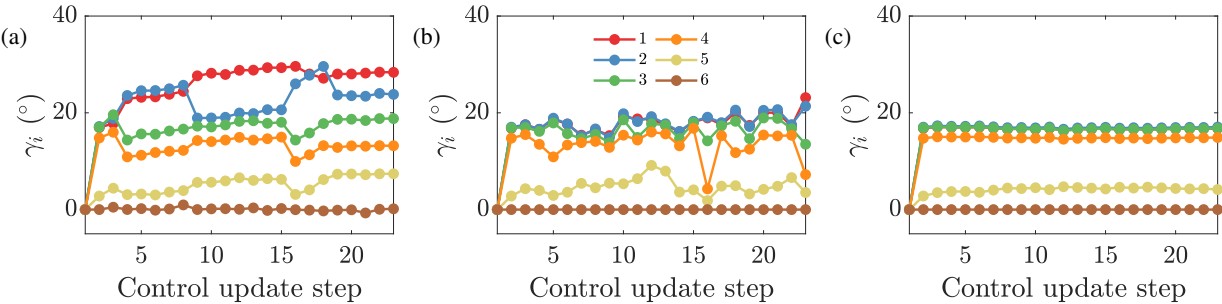

**Figure 14.** Yaw misalignment angles for each turbine in the wind farm for (a) online control using the previous optimal parameters to initialize the next state (ND1), (b) online control using the initial parameters to initialize the next state (ND2), and (c) and the static state estimation control (ND3).

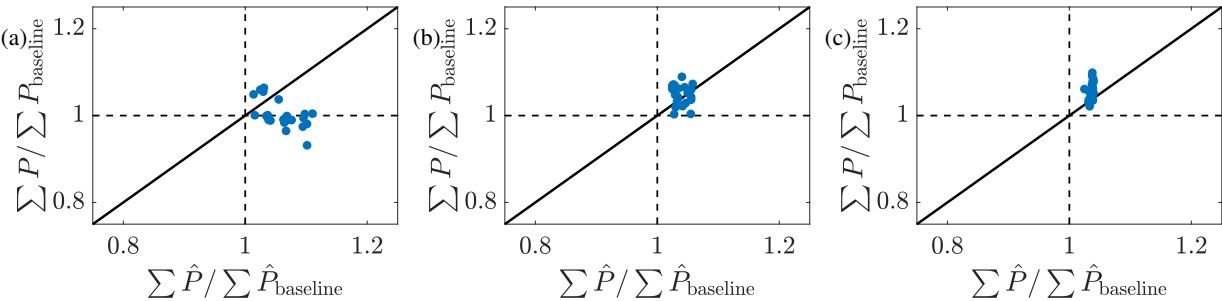

**Figure 15.** Relationship between the LES wind farm power production compared to the wake model wind farm power production prediction for (a) online control using the previous optimal parameters to initialize the next state (ND1), (b) online control using the initial parameters to initialize the next state (ND2), and (c) and the static state estimation control (ND3). The LES power production is given by $P$ and the wake model prediction is given by $\hat{P}$.

The relationship between the model predicted and LES measured power production for the three cases is shown in Figure

15. Case ND1 has an increased occurrence of wake model over-prediction of the power production while Case ND3 has an increased occurrence of wake model under-prediction. Case ND2 has approximately equal occurrence of under- and over-prediction. The efficacy of the state estimation is shown in Figure 16. Both Cases ND1 and ND2 are able to estimate the power production for the downwind turbine in the baseline, greedy operation (the first time step) and with yaw misalignment. Since Case ND3 uses static state estimation, there are some discrepancies between the LES power production and the lifting line

model (Figure 16(c)). The power production for the most upwind turbine is modeled accurately using $P_p = 3$, although the LES power production is generally slightly lower, indicating $P_p > 3$ for this ADM and ABL state.

The most successful dynamic control framework utilized in the conventionally neutral ABL is the static state estimation methodology. While the optimal yaw misalignment angles change slightly as a function of time (Figure 14), the wake model parameters are fixed. Since the flow is quasi-stationary, the wake model parameters should not change significantly as a function

of time. However, the wake model parameters may have a function dependence on $\gamma$, the yaw misalignment for the upwind



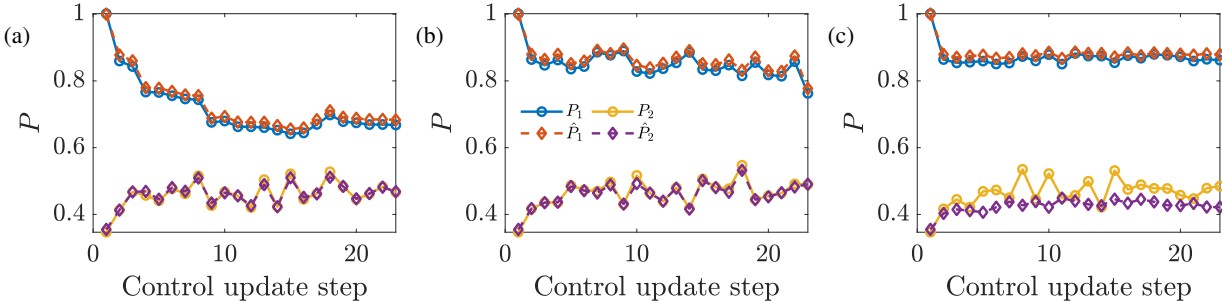

**Figure 16.** Time averaged power production for the first and second wind turbines in the wind farm as a function of the control update steps for (a) online control using the previous optimal parameters to initialize the next state (ND1), (b) online control using the initial parameters to initialize the next state (ND2), and (c) and the static state estimation control (ND3). The LES power production is given by $P$ and the wake model state estimation is given by $\hat{P}$.

turbines. This potential dependence of $k_w$ and $\sigma_0$ on yaw misalignment was not incorporated explicitly in the present modeling framework and is recommended for future work.

The static state estimation with dynamic yaw controller is able to outperform the lookup table control (Table 1). This indicates that while the wake model parameters are fixed, the optimal yaw misalignment angles differ even with changes to the mean wind direction less than $1°$. As such, the lookup table based yaw misalignment strategy is unlikely to be optimal in a general setting since it relies on wind speed and direction bins of arbitrary size. Instead, in a lookup table approach, the wake model parameters could be tabulated instead of the optimal yaw misalignment angles. Optimal yaw misalignments can be calculated dynamically, on-the-fly using the computationally efficient model described in Section 2.1 or a mid-fidelity model (e.g. WFSim, Boersma et al., 2018) could be used to compute discrete yaw angles in wind condition bins and the continuous optimal yaw function could be approximated using a neural network, for example.

### 5.3 Influence of $P_p$

The wind turbine power production as a function of the yaw misalignment in the wake model is given by Eq. 6. The parameter $P_p$ is uncertain. Following actuator disk theory, $P_p = 3$, although experiments typically show $P_p \leq 2$ for wind turbines and wind turbine models with rotation (e.g. Medici, 2005). With the ADM used presently, $P_p = 3$ should be an accurate approximation but will be imperfect since actuator disk theory applies only to one dimensional, steady flow. Since $P_p$ is wind turbine and likely site-specific, it is likely in a wake steering application that the precise value of $P_p$ is unknown *a priori*. In this section, we will model $P_p$ as 2 (ND4) and 4 (ND5) using the same control architecture as Case ND2. $P_p = 2$ will lead to an underestimate of the power production loss due to yaw misalignment and $P_p = 4$ will lead to an overestimate.

The power productions as a function of the yaw update steps for Cases ND4 and ND5 are shown in Figure 17. Case ND4 with $P_p = 2$ has 3.0% less power production than baseline greedy operation while Case ND5 with $P_p = 4$ has 5.1% more power than baseline control. Case ND2 and ND5 are significantly better than ND4. With $P_p = 2$, the model prediction for the optimal yaw misalignment angles are high, with the first three upwind turbines misaligning by almost $\gamma = 40°$ (Figure 18(a)).



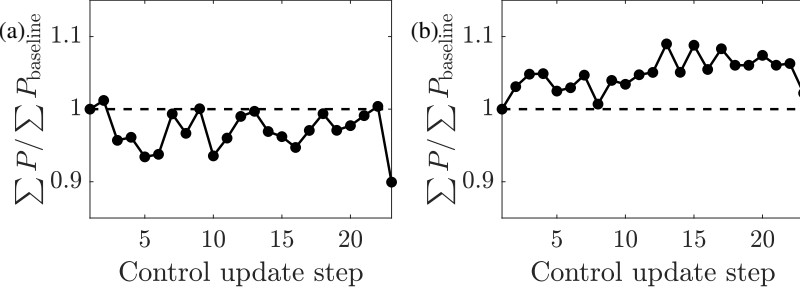

**Figure 17.** Time averaged wind farm power production as a function of the control update step for online control using the initial parameters to initialize the next state and (a) $P_p = 2$ (ND4) and (b) $P_p = 4$ (ND5). The wind farm power is normalized by the power production of the aligned wind farm case.

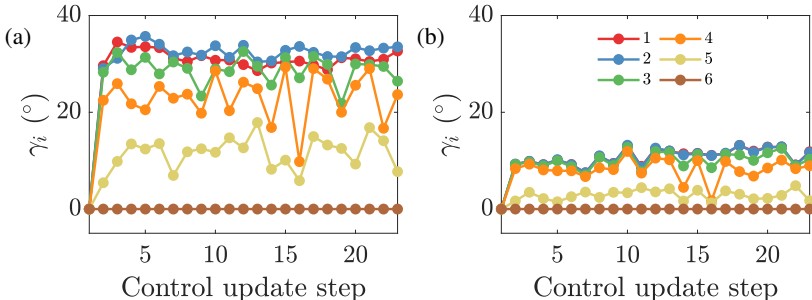

**Figure 18.** Yaw misalignment angles for each turbine in the wind farm for online control using the initial parameters to initialize the next state and (a) $P_p = 2$ (ND4) and (b) $P_p = 4$ (ND5).

With $P_p = 4$, the penalty for yaw misalignment is significant and no turbine misaligns more than $\gamma = 20°$ (Figure 18(b)). For the present conventionally neutral ABL and ADM implemented, $3 < P_p < 4$ for the leading upwind turbine. The success of
Case ND5 with $P_p = 4$ suggests that small yaw misalignments can still increase the wind farm power production significantly with respect to the baseline greedy control.

    The LES power productions and EnKF state estimated powers as a function of the yaw control updates are shown for the two $P_p$ cases in Figure 19. For Case ND4, the upwind turbine power production is significantly over-predicted. The EnKF does not estimate the state for the most upwind turbine since there are no wake model parameters which influence its production.
The power production for the second wind turbine is accurately estimated even with $P_p = 2$. This again shows that the state estimation is likely overparameterized where the EnKF is making up for the incorrect $P_p$ model by altering $k_w$ and $\sigma_0$ unphysically. The power productions and EnKF estimations for the first two wind turbines for Case ND5 show that $P_p = 4$ is a more accurate estimate than $P_p = 2$. Again, the downwind turbine power is estimated accurately with the incorrect value of $P_p$.

    The comparison between the wake model power predictions against the LES power production are shown in Figure 20.
With $P_p = 2$ (Figure 20(a)), the wake model significantly overpredicts the power production of the wind turbine array with expected power increases over the baseline of $25\%$ but a power decrease with respect to the baseline realized. On the other

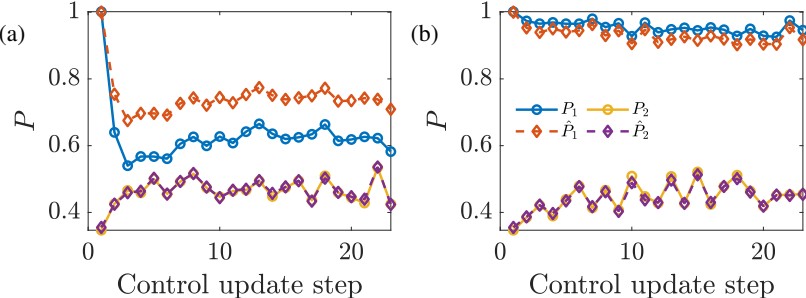

**Figure 19.** Time averaged power production for the first and second wind turbines in the wind farm as a function of the control update steps for online control using the initial parameters to initialize the next state and (a) $P_p = 2$ (ND4) and (b) $P_p = 4$ (ND5). The LES power production is given by $P$ and the wake model state estimation is given by $\hat{P}$.

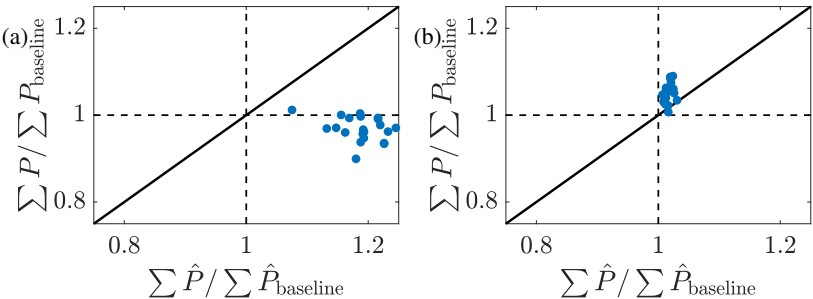

**Figure 20.** Relationship between the LES wind farm power production compared to the wake model wind farm power production prediction for online control using the initial parameters to initialize the next state and (a) $P_p = 2$ and (b) $P_p = 4$. The LES power production is given by $P$ and the wake model prediction is given by $\hat{P}$.

hand, with $P_p = 4$ (Figure 20(b)), the wake model underpredicts the power production of the wind turbine array for nearly all control update steps. Interestingly, Case ND5 outperforms Case ND2 ($P_p = 3$), but not significantly. Comparing Figures 15(b) and 20(b), it is clear that, in this simulation, the lifting line model prediction of downwind turbine power is less conservative 535 as a function of increasing $\gamma$. Therefore, the model is likely slightly over-estimating the true optimal yaw misalignment angle magnitudes when $P_p = 3$.

Overall, the sensitivity analysis on $P_p$ suggests that given a model application where $P_p$ is unknown, a conservative estimation should be taken (e.g. $P_p = 4$). With the present data-driven dynamic controller, underestimating $P_p$ leads to the wake model estimating a state which would lead to high wake losses with baseline greedy control. There is no pathway for the 540 state estimation to discern the discrepancy between an incorrect $P_p$ model or, for example, changing atmospheric conditions which are giving rise to worsening wake losses given baseline control. Future work should focus on methodologies to robustly estimate $P_p$ from SCADA data.



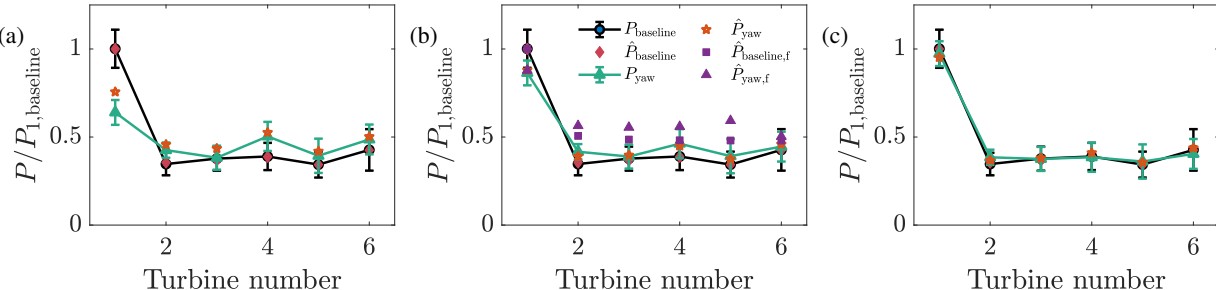

**Figure 21.** Wind turbine power production from LES $P$ and wake model $\hat{P}$. $P_{1,\text{baseline}}$ is the LES power production for the leading upwind turbine from control update step 1 where the wind farm is operated with greedy baseline control. $\hat{P}_{\text{baseline}}$ is the wake model fit to $P_{\text{baseline}}$ using EnKF estimation. $P_{\text{yaw}}$ is the LES power production for control update step 2 with yaw misalignment incorporated. $\hat{P}_{\text{yaw}}$ is the wake model prediction of $P_{\text{yaw}}$ using $k_w$ and $\sigma_0$ fit based on control update step 1 and with the optimal yaw misalignment angles which were implemented by control update step 1. (a) $P_p = 2$, (b) $P_p = 3$, and (c) $P_p = 4$. The error bars represent one standard deviation in the power data as a function of time. The subscript 'f' denotes power predictions from the FLORIS wake model (Annoni et al., 2018) with the Gaussian wake model (Bastankhah and Porté-Agel, 2014) and model parameters prescribed by Niayifar and Porté-Agel (2016).

## 5.4 Accuracy of wake model predictions

The accuracy of the wake model power predictions are assessed in this section by comparing the LES power measurements to
the wake model power predictions from the previous time step. As detailed in Section 3, the simulation is initialized with greedy
yaw alignment which is held fixed for $n_T$ time steps (control update 1), after which yaw misalignment angles are implemented
for $n_T$ steps (control update 2). The yaw angles are subsequently updated dynamically every $n_T$ simulation steps. At control
update 1, the previous $n_T$ steps of yaw aligned operation are used to compute $P_{\text{baseline}}$, the time averaged power production for
each wind turbine. $P_{\text{baseline}}$ is used to estimate $k_w$ and $\sigma_0$ using the EnKF such that $|P_{\text{baseline}} - \hat{P}_{\text{baseline}}|$ is minimized. With
the estimated model parameters, the optimal yaw misalignment angles are computed for each wind turbine. Using $k_w$ and $\sigma_0$
estimated and the optimal yaw angles computed at control update 1, $\hat{P}_{\text{yaw}}$ is predicted which is attempting to represent $P_{\text{yaw}}$,
the average power production over the $n_T$ steps following control update 1. The computation of $P_{\text{yaw}}$ is completed at control
update 2 and can be compared directly to $\hat{P}_{\text{yaw}}$ to validate the predictive capabilities of the lifting line model and the estimated
model parameters. In short, $\hat{P}_{\text{baseline}}$ represented $P_{\text{baseline}}$ and it is an estimation or fit because the model had knowledge of
$P_{\text{baseline}}$. $\hat{P}_{\text{yaw}}$ is a prediction since the model had no knowledge of $P_{\text{yaw}}$. The LES measured and wake model estimated and
predicted power productions are shown in Figure 21 for $P_p = 2$, 3, and 4.

The mean absolute error for the lifting line model power estimation was 0.0037 for all three cases since $P_p$ does not affect
the fitting with yaw aligned control enforced. The mean absolute errors for the lifting line model power predictions were 0.044,
0.015, and 0.018, given as a fraction of $P_{1,\text{baseline}}$, for $P_p = 2$, 3, and 4, respectively. The mean absolute errors as a function
of the control update steps for the three simulations are shown in Figure 22. The average over the control update steps of the
mean absolute errors for the three cases are 0.05, 0.029, and 0.036 for $P_p = 2$, 3, and 4, respectively. Qualitatively, $P_p = 3$ and
4 results in predictions which are accurate and within one standard deviation of the mean. $P_p = 2$ results in more inaccurate



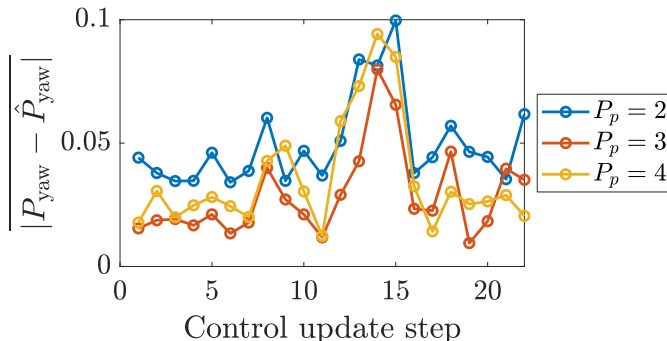

**Figure 22.** The mean absolute errors for the lifting line model predictions as a function of the control update step for the conventionally neutral ABL with $P_p = 2$, $3$, and $4$.

predictions, with elevated inaccuracy for the leading upwind turbine. Overall, these results, in tandem with the field experiment results of Howland et al. (2019), suggest that the lifting line model (Shapiro et al., 2018) provides accurate predictions of the

power production of wind farms within yaw misalignment given data-driven calibration to yaw aligned operational data.

The baseline and yaw misaligned power predictions using the FLORIS wake model package (Annoni et al., 2018) is also shown in Figure 21. The FLORIS model implementation uses the Gaussian wake model (Bastankhah and Porté-Agel, 2014) with the wake spreading rate $k^*$ approximated using the empirical LES fit between $k^*$ and the turbulence intensity given by Niayifar and Porté-Agel (2016). Since the Gaussian wake model parameters are not calibrated to the site-specific LES of this

wind farm, the inaccuracy in representing $P_{\text{baseline}}$ is expected according to the typical fidelity of engineering wake models (Stevens and Meneveau, 2017). The mean absolute error for the power production prediction in yaw misalignment averaged over the six wind turbines in the array is $0.02P_{1,\text{baseline}}$ and $0.11P_{1,\text{baseline}}$ for the lifting line model with data assimilation and the Gaussian model with an empirical wake spreading rate as a function of turbulence intensity, respectively. $P_{1,\text{baseline}}$ is the power production of the leading upwind turbine in greedy control. The EnKF data assimilation has reduced the error in the

prediction of the power production in yaw misalignment by an order of magnitude compared to *a priori* prescribed empirical model parameters. Since the greedy wake losses in FLORIS differ from the LES power production, FLORIS will also predict different yaw misalignment angles in its model-based optimization. For greenfield applications before wind farm construction, SCADA data is not available and data assimilation methods cannot be used, necessitating empirical methods such as those suggested by Niayifar and Porté-Agel (2016). For operational wind farm control optimization, site-specific data assimilation

increases the accuracy of the model predictions (Figure 21).

**5.5 Influence of the wind farm alignment**

The wake losses and potential for wake steering to increase wind turbine array power production depends on the wind turbine layout (see e.g. experiments by Bossuyt et al., 2017). In the previous section, the six wind turbines were aligned at an angle of $18°$ from the horizontal (Figure 6). The mean wind direction at hub height is approximately $15°$-$16°$ in this conventionally



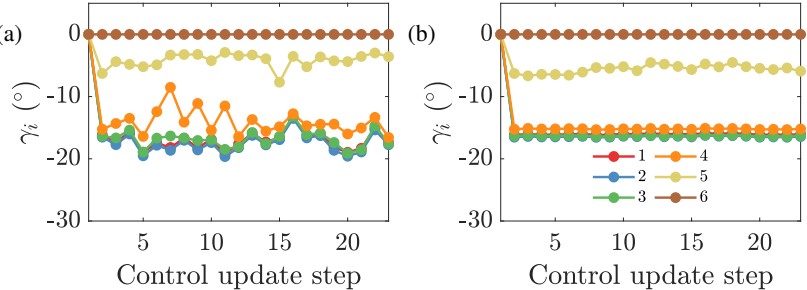

**Figure 23.** Yaw misalignment angles for each turbine in the wind farm for online control using (a) the initial parameters to initialize the next state (ND141) and (b) static state estimation parameters (ND142) for wind farm alignment at $14°$.

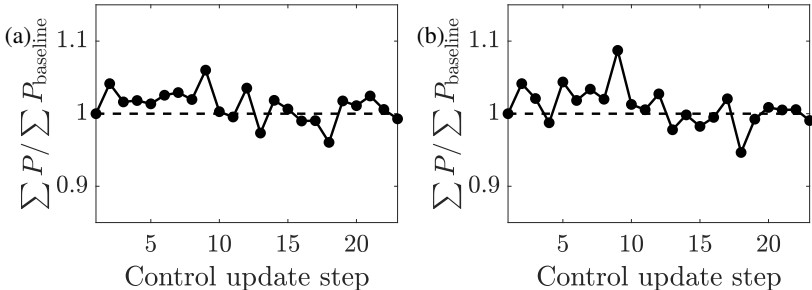

**Figure 24.** Time averaged wind farm power production as a function of the control update step for online control using (a) the initial parameters to initialize the next state (ND141) and (b) static state estimation parameters for wind farm alignment at $14°$ (ND142). The wind farm power is normalized by the power production of the aligned wind farm case.

neutral ABL. In this section, the wind turbine column alignment is changed to $14°$ from the horizontal and the array is embedded within the same conventionally neutral ABL. As a result of this array alignment, the optimal yaw misalignment angles will change from positive (counter-clockwise rotation viewed from above) to negative (clockwise). It should be noted that this sensitivity analysis is not a controlled experiment to test the benefit of yawing in opposite directions since asymmetries exist in the conventionally neutral ABL as a result of the veer angle and the magnitude of partial waking is not held fixed between
the two layouts.

    For the wind turbine array aligned at $14°$, the dynamic wake steering controller is tested with dynamic (ND141) and static state estimation (ND142). With a wind farm alignment along $14°$ and the mean wind direction at hub height of approximately $15°$-$16°$, the optimal yaw misalignment angles are negative (clockwise viewed from above). The yaw misalignment angles implemented as a function of the control update steps are shown in Figure 23 for dynamic and static state estimation architec-
tures. The qualitative magnitude of the yaw misalignment angles are similar to the angles selected for the $18°$ alignment case (Section 5.1).

    The power productions for the two wake steering controllers are shown in Figure 24. The temporally averaged power production increase over baseline, greedy operation is $1.1\%$ and $1.0\%$ for the dynamic and static state estimation cases, respectively. There is no significant difference in the mean power production between these two state estimation methodologies for this





wind farm alignment (see Table 1). Further, neither wake steering control case increases power significantly over greedy control. While the power production increase over the greedy control is less for the $14°$ case with negative yaw misalignment than for the $18°$ case with positive yaw misalignment this is not a controlled experiment since the degree of partial waking is different between the two cases. The wind farm has more direct wake interactions, with less partial waking, for the $14°$ alignment as evidenced by the lower power production in greedy control (Table 1). Previous simulations have shown that for

a controlled experiment of direct wind farm alignment, positive yaw misalignment (counter-clockwise) is superior to negative yaw misalignment (clockwise) (see e.g. Fleming et al., 2015; Miao et al., 2016), although this will depend on the specific ABL and wind farm layout simulated. Archer and Vasel-Be-Hagh (2019) proposed that this difference is a function of Coriolis forces in the ABL, although future work should quantify the effect of latitude and hemisphere locations as well as the influence of non-traditional effects (Howland et al., 2020a). The degree of power production increase as a result of wake steering is a strong

function of the wind farm alignment with respect to the wind direction at hub height, the turbine spacing, the shear, and veer. The present simulations reveal that it is reasonable to capture increases in power production with negative (clockwise) wake steering even with a wind turbine model with $P_p \approx 3$.

## 6   Conclusions

A suite of large eddy simulations has been performed to characterize the performance of a dynamic, closed-loop wake steering

wind farm control strategy. The controller was designed for the application of real-time utility-scale wind farm control based on only SCADA data without requiring a LiDAR on site. The physics- and data-driven ensemble Kalman filter and wake model based controller was validated in uniform inflow LES before being tested in conventionally neutral ABL conditions with Coriolis, shear, and veer. The analytic gradient ascent optimal yaw selection allows for real-time dynamic wind farm control. The sensitivity of the power production increase via wake steering over greedy, yaw aligned control was characterized

as a function of the controller architecture, $P_p$, state estimation architecture, wind farm layout, and ABL conditions.

Within the quasi-stationary conventionally neutral ABL, the optimal yaw misalignment angles do not change significantly with time. Within this simplified ABL environment, a static wake steering strategy, where the yaw misalignments do not change, increased power production by $5.4\%$ with respect to baseline greedy control. Dynamic wake steering with dynamic state estimation increased power production by $4.6\%$, slightly less than the static yaw misalignment strategy but not significantly.

The highest power production occurred with a wake steering strategy where the model parameters were fixed and the state estimation was not performed every control update step but the yaw misalignment angles were updated according to the local wind conditions. This result indicates that in a lookup table wake steering approach, the wake model parameters should be tabulated and the yaw angles should be calculated on-the-fly given exact local wind conditions, rather than direct optimal yaw misalignment angle tabulation. All three of these wake steering cases increased power significantly over greedy, aligned control

although the differences between the three control architectures were not significant.

The importance of the model for individual wind turbine power production degradation as a function of the yaw misalignment angle, and in particular $P_p$, was demonstrated where $P_p = 3$ or $4$ lead to an increase in power production with respect to



greedy operation while $P_p = 2$ lead to a loss in power. Wake steering cases with $P_p = 3$ and $4$ led to a significant increase in power production compared to greedy control while $P_p = 2$ did not. Since $P_p$ depends on the wind turbine model and ABL characteristics and there is no accepted general framework for determining $P_p$, this should be investigated in future work. With $P_p = 3$, the wake model makes accurate forecasts of the power production over a future time horizon given the yaw misalignment strategy that is implemented. This accuracy gives confidence to the data-driven EnKF state estimation and lifting line wake model for the application of wake steering control. The combined lifting line model and EnKF state estimation has an order of magnitude reduced predictive error than the Gaussian wake model with an empirical wake spreading rate in this conventionally neutral ABL simulation.

The results are qualitatively similar when a wind farm of different alignment is embedded within the conventionally neutral ABL. The power production is decreased with a wind farm alignment of $14°$ compared to $18°$ and with a clockwise yaw misalignment compared to a counter-clockwise, although this was not a controlled experiment of the influence of the direction of yaw misalignment.

While the conventionally neutral ABL cases were not designed to model a specific wind farm and to compare to field data, this LES testbed paradigm is useful for the rapid prototyping of optimal wind farm control architectures. The main purpose of this study was predominantly to establish the dynamic wake steering framework and perform sensitivity analysis on the controller architecture rather than the ABL or LES setup. The uncertainties and sensitivities in this study associated with the wall model, subfilter scale model, wind turbine model, and ABL characteristics such as boundary layer inversion height were not investigated in detail and are left for future work. More reliable and generalizable estimates for $P_p$ (Liew et al., 2019), or generally $C_p$ as a function of $\gamma$, should be investigated. Future work should also investigate the influence of latitude and geostrophic wind direction on wake steering control performance (Howland et al., 2020a). Finally, the controller should be tested using other LES codes and in field experiments to assess the generalization of the results. Part 2 of this study will implement the dynamic optimal controller in transient ABL conditions such as the stable ABL and the diurnal cycle.

*Code and data availability.* The code is open-source and available at https://github.com/FPAL-Stanford-University/PadeOps. The GitHub repository branch for incompressible wind farm simulations is 'igridSGS.' The data will be open-access and published on the Stanford Digital Repository (https://sdr.stanford.edu/) upon publication.

## Appendix A:  EnKF test model problem

The state estimation EnKF algorithm and implementation is tested using a six wind turbine model wind farm with artificial data. Six 1.8 MW Vestas V80 wind turbines are modeled with incoming wind speed of $u_\infty = 7.5$ m/s. The turbines are spaced $6D$ apart in the streamwise direction and are directly aligned in the spanwise direction as shown in Figure A1(a). The parameters selected for the EnKF algorithm are $\sigma_{k_w} = 0.001$, $\sigma_{\sigma_0} = 0.001$, and $\sigma_P = 0.1$. The initial wake model parameters were selected as $k_w = 0.1$ and $\sigma_0 = 0.35$ for each wind turbine in the array. The model is run with a specified, artificial mean



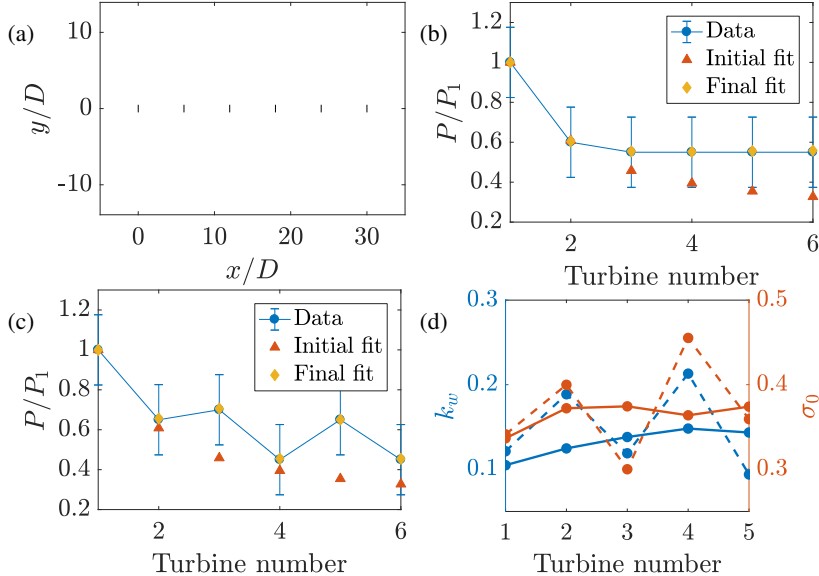

**Figure A1.** (a) Model problem setup. (b,c) The EnKF model fits for the model problem with two prescribed, artificial mean power profiles and Gaussian random noise. (d) Model parameters shown with solid (b) and dashed (c) lines. The wake model parameters for the last turbine downwind are not shown since they do not impact the state estimation accuracy.

power production profile with Gaussian random noise superposed. The model is run over 1000 model time step iterations with
$N_e = 100$. The initial and final model calibrations are shown in Figure A1(b,c). The EnKF combined with the lifting line model is able to fit the artificial wind farm data with sufficient accuracy for two different power production profiles.

As shown in Figure A1(b,c), the EnKF state estimation combined with the lifting line model are able to reproduce the power production for the artificial data to high accuracy. The ability for a one or two parameter analytic wake model to capture arbitrarily generated power production profiles should be investigated in future studies as the model may enforce unrealistic
model parameters to represent neglected physics (Schreiber et al., 2019). The validity of this data-driven framework is validated in the LES test cases in a comparison between model power predictions and LES power measurements (Section 5).

**Appendix B:  Extended conventionally neutral simulation**

The conventionally neutral ABL Case ND2 is run for 50 control update steps and the results are shown in Figure B1. The controller does not become unstable as a function of time and the magnitude of yaw misalignment angles are approximately
constant.





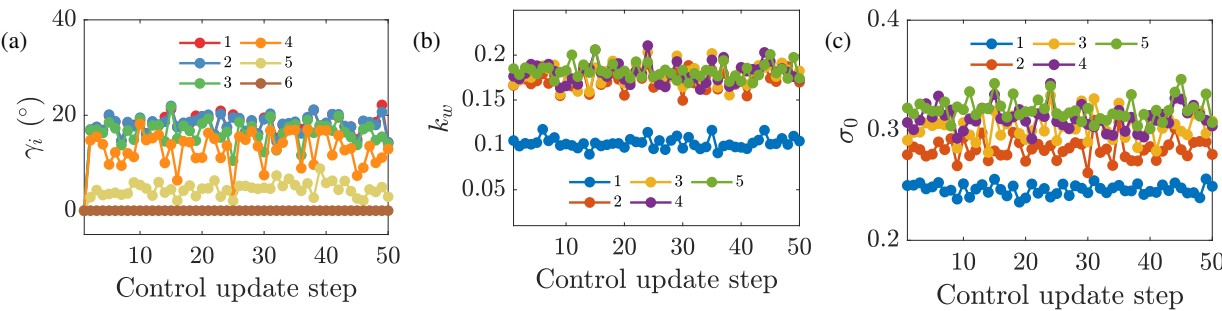

**Figure B1.** Wind farm (a) yaw misalignment angles, (b) $k_w$, and (c) $\sigma_0$ as a function of the control update steps for the extended ND2 case.

## Appendix C: Influence of local atmospheric conditions on wind turbine array power production

The quantification of the influence of new control methods on the wind farm power production is challenging in an experimental setting. In a computational environment, simulations with identical initial conditions and fixed time stepping schemes can be used to quantify the influence of the operational modifications in a controlled experiment. In a field experiment environment,
since wind conditions are constantly changing and are not repeatable due to the nature of atmospheric flows, this quantification is more challenging. Complex terrain and differences in the manufacturing and operation of turbines in standard control leads to substantial discrepancies in the instantaneous power production of freestream turbines at wind farms. Therefore, comparing yaw misaligned columns of turbines to yaw aligned leads to uncertainty in analysis. Further, conditional averages based on wind speed, direction, turbulence intensity, and atmospheric conditions may not sufficiently capture the potential physical
mechanisms which influence power production. To quantify this impact in the present simulations, the power productions as a function of the control update steps can be compared to the first control update step in the quasi-stationary conventionally neutral ABL flow. Inertial oscillations, turbulence, and sampling error will cause discrepancies between the first and subsequent control update steps even with the yaw aligned control strategy held fixed in the quasi-stationary flow. The average power production compared to the first yaw control update step is $4.3\%$ and $9.0\%$ higher for the yaw aligned (Case NA) and
dynamic closed-loop control (Case ND2), respectively. The increase observed in Case NA indicates that the simulation had not completely converged to the quasi-stationary state upon control initialization although this does not affect the qualitative conclusions of Section 5. The true increase in power production due to wake steering in Case ND2 compared to Case NA is $4.6\%$ over the same simulation temporal window. These results highlight the need to develop robust statistical methods to analyze the impact of changing wind farm control strategies compared to the baseline.

*Author contributions.* M.F.H., S.K.L., and J.O.D. conceived the work. A.S.G. and M.F.H. developed the LES code. M.F.H. conducted analysis. M.F.H. wrote the manuscript. All authors contributed to edits.



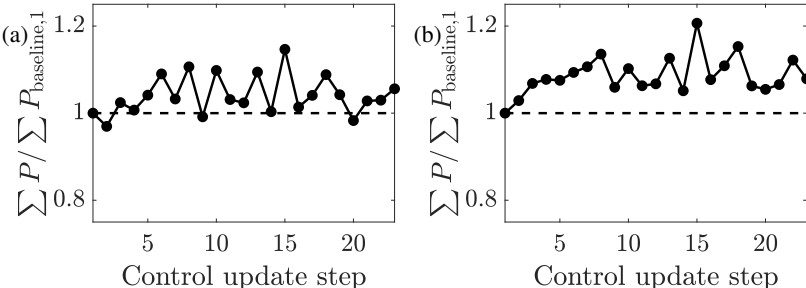

**Figure C1.** Time averaged power production as a function of time normalized by the time averaged power production of the first control update step for yaw aligned greedy control. (a) Yaw aligned greedy control (Case NA) and (b) closed-loop dynamic control (Case ND2).

*Competing interests.* The authors declare no conflicts of interest.

*Acknowledgements.* M.F.H. is funded through a National Science Foundation Graduate Research Fellowship under Grant No. DGE-1656518 and a Stanford Graduate Fellowship. A.S.G. was funded by Tomkat Center for Sustainable Energy at Stanford University. S.K.L. acknowl-

edges partial support from NSF-CBET-1803378. All simulations were performed on Stampede2 supercomputer under the XSEDE project ATM170028.



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
