# Peer review of "Optimal closed-loop wake steering, Part 1: Conventionally neutral atmospheric boundary layer conditions"

_Wind Energy Science, 2020_

## Referee Comment (RC1) · Anonymous Referee #1 · 30 Apr 2020

Review comments for "Optimal closed-loop wake steering, Part 1: Conventionally neutral atmospheric boundary layer conditions" by Howland et al. in Wind Energy Science, 2020.

Overall: The article presents a framework for closed-loop wind farm control, particularly focused on wake steering. The framework consists of a lifting line wake model and an ensemble Kalman filter, where the wake model is continuously calibrated to previous results, and used to optimize the wind farm control settings. The framework is tested using LES and the sensitivity of various parameters is examined.

The topic is interesting and important. However, the article is quite long, and certain

places it seems unfocused and occasionally self-contradictory, or even biased. Sensitivity studies on model parameters are generally very important, but it seems that the application of the wake model itself within the framework is the main drawback here. I recommend major revisions.

General comments: Given that the articles requires major revisions, I will only provide major comments, which are lengthy enough.

1. Wake model. Overall, the inclusion of the wake model seems to be the weak, but predominant, part of the article. That is not a criticism of the wake model itself. The lifting line wake model definitely seems to have merit, and engineering models are important. It is fully understandable that model developers wish to spread their model and test their capabilities. However, the present application should be better addressed and motivated. Does the analysis actually strengthen or weakening the general use of the wake model? Does the inclusion of the wake model improve the potential for closed-loop wind farm control, or are the results so model dependent that one should rather use a more model-free approach?

It seems that the calibration of model parameters frequently ends up compensating for the missing physics, which affects several of my following points. This is also one of the major self-contradictions within the article. On one hand, the authors repeatedly write that the model is "physics-based", which in isolation is correct. However, the current application of the model also includes several other models/assumptions, which in combination is questionable. It also leads the authors to give statements like: - "the two parameter lifting line model may be overparameterized, which leads to overfitting." - "The ability of a one or two parameter analytical wake model ....may enforce unrealistic model parameters to represent neglected physics.". Clearly, the model does not capture all the physics. Hence, the desirable physics of the original model appears to become a limitation, rather than an benefit, when various additional models and assumptions are combined, see below. The results and the aforementioned quotes indicate these concerns, but it is not reflected in the abstract nor in the conclusion, which

hence seems somewhat selective.

I will try to elaborate on a number of points, where this seems to have a significant impact.

a) Model Description and Assumptions: - The assumption of thrust following cos^2 gamma could potentially have a big impact on the results. The article investigates PP, but this seems to be an equally important assumption and the impact should not be neglected. The article states "Future work should focus on methodologies to robustly estimate PP from SCADA data". That's in principle good, but I question the statement "there is no accepted framework for determining PP"(line 635). It seems that it could relatively easily to perform simulations using an actuator disc model or an aero-elastic tool, e.g. FAST, to fit both PP and the coefficient for CT as function of yaw for the specific turbine. This would render Section 5.3 obsolete as it stands now. The sensitivity to PP is obviously interesting, but it would seem more appropriate to address it using your reference to Liew et al. as also commented in line 650.

- Linear superposition: This is a highly questionable assumption, which will also affect the results in terms of the overfitting and parameter compensating for lack of physics. An improved wake superposition has recently been proposed by Zong and Porté-Agel, 2020, which is physically consistent and shows improved results, also for wake steering.

- Please define the "effective velocity"(line 136). Does that correspond to rotor averaged or based on the power production?

b) Advection time: This section could be significantly reduced and rephrased. Most of the explanation deals with Taylors hypothesis, but at the end the advection time is adjusted to be twice this "to account for errors associated with the simple advection model". This appears as a somewhat random choice in the current context. The proper explanation could/should involve (conservative) estimates of the wake propagation velocity, i.e. the wake propagates slower than the mean flow. There are numerous

references for this.

2. Introduction. Although a good review is appreciated, it could be shortened and more focused. The section on derating seems too long for an article focusing on wake steering. Wind farm control is a very active field, so new articles are constantly being publish. Hence, I will just recommend the inclusion the recent review by Kheirabadi and Nagamune, 2019, which also quantifies the potentials of wind farm control. As also concluded by Kheirabadi and Nagamune, wake steering has been shown to have the largest potential, but their Figure 4 also reveal that power increase from wake steering is not guaranteed. Hence, some statements could be less assured, e.g. line 116 in the article.

3. Steady vs dynamics: Generally, it seems "unfair" to use a steady state model in a dynamic framework, although one can argue if 30min is even dynamic. And by "unfair", I mean on the analytical model. How would the performance be if it was applied on a more realistic control scenario of 5-10min? I suspect this is given in Part 2, but this should be a self-contained article.

There are several places, where the wording appears inconsistent. This is particular the case in Section 4, where "the flow is stationary" is used several times and at the same time "the dynamics of the wake" and "counter rotating vortices". Such statements appear contradictory. LES simulations of the flow behind a disc/rotor should not be stationary, even with uniform inflow. The wake will be dynamic (as you mention), e.g. meandering. This is also seen in Figure 5. There it appears as if the two turbines are positioned so close that the first wake doesn't start to meander/breakdown, but the second does. Could the quasi-steady behavior be caused by the coarse resolution of 5-6 cells per disc?

Several question arise from Section 4: - Is uniform inflow a good and representative case? Does it truly provide validation of optimization, because it turns the turbine in the right direction? - Line 285 states that AD is good for far wake. But is 4D far wake

in uniform inflow? A quick back-of-the-envelope assessment for where the far wake start can be done using the engineering expression in Sørensen et al., 2015. For a 0.01% inflow turbulence and propagation velocity of 0.7, I get an estimate of 13-14D downstream, before it is fully transitioned to far wake with a Gaussian velocity deficit, which is another of the model assumptions. What is the implication of this assumption? - Please rephrase for clarification line 377: "reduced in size and intensity". What is reduced in size and what intensity? The deficit? It would be difficult(=impossible) to see from an instantaneous velocity contour. Similarly, it is not possible to see how the figure "indicating that for larger columns of turbines the potential for power increases due to wake steering are larger" due to the dynamics of the instantaneous flow.

4. Quantification. Several issues relating to quantification could be improved and clarified. - Are the power productions in Table 1 normally distribution? Otherwise, the standard deviation could be biased and misleading. - The inclusion of whether results are significant can only be applauded. Line 434 defines "significant superior...if the mean array power production ....is more than one standard deviation larger than the other". However, the use is biased, because the analysis is essentially only one-sided. The authors only compare controlled mean power - 1 standard deviation to the mean power of the baseline. The authors should also include the standard deviation of the baseline, because it's two overlapping distributions. That essentially means none of the cases are significantly superior if using the authors definition of statistical significance.

5. Optimal yaw angles The determined optimal yaw angles seem quite large, although others report similar values. In figures 8 and 14 the yaw angles of the first 3-4 turbines are 15-20deg. The following comments concern additional required information, because it is difficult to assess from the article in its current form.

a) Zong and Porté-Agel also shows the secondary steering effect, which other recent articles are also investigating, see e.g. King et al., 2020. The secondary steering effect essentially implies that the next wake will also be partly steered for "free"(without power loss). The Liew et al. 2020 reference states that McKay et al. 2013, Bartl et

al. 2018, and Hulsman et al.2020, all show that the local inflow direction changes, and the waked (2nd) turbine should adjust to the inflow. Hence, it would indicate that if the second turbine also yaws 20deg, it actually corresponds to a larger relative yaw angle. So why would the optimization give the same(or relatively larger) yaw angles for the first 3 turbines? Should the yaw angles not be reduced at least by the secondary effects or is this not capture by the LES?

b ) A quick estimate of the power yaw loss with PP = 3, yields the following table:

yaw CP

0 1.0000

5.0000 0.9886

10.0000 0.9551

15.0000 0.9012

20.0000 0.8298

25.0000 0.7444

30.0000 0.6495

That means the first turbine yawing 15-20deg looses approx. 10-17% of the power compared to its own baseline(also shown in Figure 16). Line 480 states "power production penalty...is significant beyond 40deg". The 40deg seems "random", as one could just as well argue that losses of 10-17% are significant. In order for the entire farm to produce more, this loss should be recovered by the next turbine. However, the next 3 turbines experience similar losses. Therefore, increase in velocity have to compensate for all these turbine losses. If I use estimates from Figure 16, I read an initial power production of approx. 35% and an optimal power production of approx. 50% for the second turbine relative to the first. Hard to see from this rough estimate if the second turbine actually gain more than what the first turbine losses. Archer and

Vasel-Be-Hagh also concluded there needs to be at least 2 turbines downstream a yawed turbine to regain the power loss.

c) In order to recover the power loss, the wake velocity have to increase by V2/V1 = (0.50/0.35 * CP0/CPyaw)^(1/3) where CP0 corresponds to my previous table of 0deg yaw, and CPyaw is the remaining. Hence, the velocity increase(V2/V1) on the second turbine needs to be:

yaw V2/V1

0 1.1262

5.0000 1.1305

10.0000 1.1436

15.0000 1.1660

20.0000 1.1985

25.0000 1.2427

30.0000 1.3005

So a 17-20% increase in mean velocity. It would be nice to see some kind of velocity plots, either contour plots before and after control to verify this. Therefore, it would also be nice to see the actual power production of the individual turbines compared to baseline for the various scenarios, because it's difficult to assess which turbines actually make up for the losses and increase the total production in the accumulated plots, e.g. Figure 9. The authors have attempted to do so in Figure 21, but it's very difficult to distinguish. It seems turbines 2, 4 and 5 recover the losses for PP = 3? Perhaps use bar plots with clearer standard deviations?

Shorter comments: - The inclusion of FLORIS (Section 5.4) seems superfluous. I suggest to leave it out to keep the analysis more focused, and shorter. - The definition

of 0deg north seems redundant. Where is it used? It seems to be in conflict with Figure 6, as the wind direction with 0deg north would not be 16deg, but a 254deg. - Terminology might change over time and for different researchers. Some use kidney shaped, others curled wake. However, the vortex pair has been observed in the wake behind a yawed turbine prior to 2016 (line 131). The earliest reference I could find was Mikkelsen, 2004, see Section 8.3. - Please introduce/explain Table 1 more when it is first presented. - Please provide a proper description of the turbine. Line 398: "selections based on NREL 5MW" does not seem reproducible by other scientists. - Details are difficult to see in many of the figures. For instance, reduce the axis limits on Figure 15 and similar. Figure 21 is very hard to distinguish.

Additional references:

1. Kheirabadi, Ali C. and Nagamune, Ryozo, 2019, A quantitative review of wind farm control with the objective of wind farm power maximization. Journal of Wind Engineering and Industrial Aerodynamics.

2. Zong, Haohua and Porté-Agel, Fernando, 2020, A momentum-conserving wake superposition method for wind farm power prediction, Journal of Fluid Mechanics

3. Mikkelsen, R. F., 2004. Actuator Disc Methods Applied to Wind Turbines. PhD, Technical University of Denmark

4. Sørensen, J. N., Mikkelsen, R. F., Henningson, D. S., Ivanell, S., Sarmast, S., and Andersen, S. J., 2015. Simulation of wind turbine wakes using the actuator line technique. Philosophical Transactions of the Royal Society A: Mathematical, Physical and Engineering Sciences

5. King, J., Fleming, P., King, R., Martínez-Tossas, L. A., Bay, C. J., Mudafort, R., and Simley, E.: Controls-Oriented Model for Secondary Effects of Wake Steering, Wind Energ. Sci. Discuss., https://doi.org/10.5194/wes-2020-3, in review, 2020.

---

## Author Comment (AC1) · 11 May 2020

**Response to Referee #1**

We thank the referee for their review and their thoughtful comments. Point-to-point responses can be found below, and the relevant changes will be made to the manuscript during the revised submission stage (https://www.wind-energy-science.net/peer_review/interactive_review_process.html), colored in red.

**Comment #1**

Wake model. Overall, the inclusion of the wake model seems to be the weak, but predominant, part of the article. That is not a criticism of the wake model itself. The lifting line wake model definitely seems to have merit, and engineering models are important. It is fully understandable that model developers wish to spread their model and test their capabilities. However, the present application should be better addressed and motivated. Does the analysis actually strengthen or weakening the general use of the wake model? Does the inclusion of the wake model improve the potential for closed-loop wind farm control, or are the results so model dependent that one should rather use a more model-free approach?

**Response**

Thank you for this comment. We will improve the motivation of the wake model in the manuscript and we will briefly summarize that justification and rationale here. In general, model-free wind farm power maximization has significantly slower rates of convergence than model-based optimization and may therefore be less well suited for closed-loop control in a practical, transient wind farm setting (see recent discussion by [1,2]). Model-free formulations have not been vigorously tested in transient simulation mean conditions aside from the recent paper by Ciri et al (2019) [3].

Given computational cost considerations for real-time control (i.e. computational limitations at an operational wind farm), a focus of the wind farm controls community has been to leverage existing, computationally efficient steady-state wake models for closed-loop control (see, e.g., review by Doekemeijer, Fleming & Willem van Wingerden (2019) [1]).

As the referee has suggested, an overarching objective of the submitted manuscript is to examine the sensitivity of closed-loop wind farm power maximization control to common wake model parameters and assumptions. More specifically, the assumptions invoked in the wake model presented here are functionally similar to assumptions in the popular FLORIS model software. The results in this manuscript suggest that wake models do lead to beneficial power production increases for turbine arrays, provided that the parameters in the wake models can be accurately estimated *a priori* or in a closed-loop fashion.

While model-free wind farm control is an interesting line of research moving forward, generalizations of such approaches to transient atmospheric conditions require further rigorous testing. Future work in this community (as recently discussed at the IEA Topical Expert Meeting on Wind Farm Controls) should establish standard test problems and benchmarks for the comparison of various optimization formulations such as the presently proposed methodology and model-free formulations.

**Comment #2**
It seems that the calibration of model parameters frequently ends up compensating for the missing physics, which affects several of my following points. This is also one of the major self-contradictions within the article. On one hand, the authors repeatedly write that the model is "physics-based", which in isolation is correct. However, the current application of the model also includes several other models/assumptions, which in combination is questionable. It also leads the authors to give statements like:
- "the two parameter lifting line model may be overparameterized, which leads to overfitting."
- "The ability of a one or two parameter analytical wake model ....may enforce unreal-istic model parameters to represent neglected physics.".
Clearly, the model does not capture all the physics. Hence, the desirable physics of the original model appears to become a limitation, rather than an benefit, when various additional models and assumptions are combined, see below. The results and the aforementioned quotes in-dicate these concerns, but it is not reflected in the abstract nor in the conclusion, which hence seems somewhat selective. I will try to elaborate on a number of points, where this seems to have a significant impact.

**Response**
Thank you for this thorough comment. We will update the abstract and conclusions to further reflect the sensitivity analyses which were a major component of this research.
Due to the underlying assumptions of wake models and yaw misalignment wake deflection models, these methods cannot be expected to capture all relevant physics for full-scale wind farm operation. In fact, the use of state and parameter estimation methodologies to correct unresolved physics or imperfect model assumptions have been commonly used for wind farm controls applications [see e.g. 4, 5].
Within the current closed-loop framework, the goal of the EnKF model parameter estimation is to accurately fit to a current timestep and then to apply those parameters for optimization only for the very next timestep. While it is reasonable to question the generalization of the site-specific fitting, the purpose here is not to create general parameterizations of the model parameters to be used for other conditions or wind farms but to only apply the parameters from one control step to the next. This objective and fitting consequence is thoroughly discussed in the text of this article (and will be detailed in the abstract and conclusion of the updated manuscript as per the referees comments).
More importantly, the *predictive* ability of this approach was tested here in Figures 21 and 22. As discussed in the manuscript *Section 5.4,* the present wake model approach with EnKF state estimation significantly outperforms the predictive capabilities of the commonly used Gaussian wake model with pre-defined empirical parameters with 5x lower predictive error. Therefore, with the site-specific fitting, the EnKF estimation of the model parameters significantly improves the model predictive ability. The improvement in the predictive ability of wake models with optimized model parameters is also shown in the very recent paper [2] (published online April 29, 2020).
Future work will focus on the constraining of the state estimation methodology to reduce model flexibility to test whether this improves the predictive success.

**Comment #2(a)(1)**

Model Description and Assumptions:
- The assumption of thrust following cosˆ2gamma could potentially have a big impact on the results. The article investigates PP ,but this seems to be an equally important assumption and the impact should not be neglected.

**Response**
The authors agree that the wake model assumption of thrust following cosˆ2(gamma) could also have an influence on the closed-loop control performance. This was not tested in this study since the actuator disk model used directly enforces the thrust to follow cosˆ2(gamma) and since this assumption is enforced in the lifting line model derivation [6]. In general, the thrust would follow cosˆTp(gamma), where Tp is likely less than 2 (see e.g. Bastankhah and Porte-Agel (2016) [7]). In this event, the closed-loop model parameter estimation would likely compensate by predicting smaller wake expansion coefficients (which leads to stronger wakes) than the true value given the model expectation of less streamwise thrust than is imposed in reality. We will update the manuscript with further discussion about this wake model assumption.

**Comment #2(a)(2)**
The article states "Future work should focus on methodologies to robustly estimate PP from SCADA data". That's in principle good, but I question the statement"there is no accepted framework for determining PP"(line 635). It seems that it could relatively easily to perform simulations using an actuator disc model or an aero-elastictool, e.g. FAST, to fit both PP and the coefficient for CT as function of yaw for the specific turbine. This would render Section 5.3 obsolete as it stands now. The sensitivity to PP is obviously interesting, but it would seem more appropriate to address it using your reference to Liew et al. as also commented in line 650.

**Response**
As discussed by Liew *et al.* (2020) [8], both actuator disk theory and blade element momentum theory (e.g. FAST) return Pp=3. Wind tunnel experiments [e.g. 9] and large eddy simulations [e.g. 10] have shown that Pp is more typically between 1.5-2.
Further, Pp is likely turbine specific (see e.g. Liew *et al.* (2020) [8]) and has functional dependence on shear, veer, atmospheric stability, wake impingement, and likely other parameters. Field experiments (on-going by the authors) to robustly calculate Pp require weeks or months of experimentation due to the functional dependencies mentioned above and are therefore expensive and time consuming. Therefore, the authors believe that the quantification of the influence of Pp under model parameter uncertainty are critical to future wake steering deployments.

**Comment #2(a)(3)**
- Linear superposition: This is a highly questionable assumption, which will also affect the results in terms of the overfitting and parameter compensating for lack of physics. An improved wake superposition has recently been proposed by Zong and Porté-Agel,2020, which is physically consistent and shows improved results, also for wake steering

**Response**

While the linear wake superposition method (or sum-or-squares superposition commonly used in FLORIS) are often used, the referee is correct that their physical justification is challenging. Zong and Porte-Agel (2020) (which was published after this article was submitted for review) have thoroughly addressed this long-standing open-question in the literature. The method of Zong and Porte-Agel (2020) leverages an iterative approach and assumes an empirical, pre-defined prescription of the wake model parameters which differs from the present model approach. Future work will incorporate the new superposition methodology with optimal parameter estimation.

**Comment #2(a)(4)**
- Please define the "effective velocity" (line 136). Does that correspond to rotor averaged or based on the power production?

**Response**
The effective velocity is the rotor averaged velocity, we will adjust the language in the manuscript to clarify this definition.

**Comment #2(b)**
Advection time: This section could be significantly reduced and rephrased. Most of the explanation deals with Taylors hypothesis, but at the end the advection time is adjusted to be twice this "to account for errors associated with the simple advec-tion model". This appears as a somewhat random choice in the current context. The proper explanation could/should involve (conservative) estimates of the wake propagation velocity, i.e. the wake propagates slower than the mean flow. There are numerous references for this.

**Response**
Thank you for this comment. We will rephrase this section for brevity and include additional wake propagation references in the manuscript. The invocation of twice the Taylor's hypothesis advection time was to have a strongly conservative estimate. Since the flow is quasi-statistically stationary (i.e. statistically stationary except for inertial oscillations which occur on Coriolis time scales), conservative estimates for the advection time and time-averaging of statistics have been taken. The influence of the advection time scale assumption was tested in the initial manuscript where it was shown that it did not play a significant role in the power production output of the closed-loop control, and the results are given in Table 1. Part 2, which studies the transient diurnal cycle, will investigate these times scales in more detail.

**Comment #3**
Introduction. Although a good review is appreciated, it could be shortened and more focused. The section on derating seems too long for an article focusing on wake steering. Wind farm control is a very active field, so new articles are constantly being publish. Hence, I will just recommend the inclusion the recent review by Kheirabadiand Nagamune, 2019, which also quantifies the potentials of wind farm control. As also concluded by Kheirabadi and Nagamune, wake steering has been shown to have the largest potential, but their Figure 4 also reveal that power increase from wake steering is not guaranteed. Hence, some statements could be less assured, e.g. line 116 in the article.

**Response**
We will rephrase the introduction and streamline the discussion. We will include the reference to the review paper the referee has noted.

**Comment #4**
Steady vs dynamics: Generally, it seems "unfair" to use a steady state model in a dynamic framework, although one can argue if 30min is even dynamic. And by "unfair", I mean on the analytical model. How would the performance be if it was applied on a more realistic control scenario of 5-10min? I suspect this is given in Part 2, but this should be a self-contained article. There are several places, where the wording appears inconsistent. This is particular the case in Section 4, where "the flow is stationary" is used several times and at the same time "the dynamics of the wake" and "counter rotating vortices". Such statements appear contradictory. LES simulations of the flow behind a disc/rotor should not be stationary, even with uniform inflow. The wake will be dynamic (as you mention), e.g.meandering. This is also seen in Figure 5. There it appears as if the two turbines are positioned so close that the first wake doesn't start to meander/breakdown, but the second does. Could the quasi-steady behavior be caused by the coarse resolution of 5-6 cells per disc?

**Response**
Thank you for noting this language confusion. The statements should read "statistically stationary" rather than just "stationary" and we will modify the manuscript to improve clarity and we apologize for the confusion this language has caused.
The starting point for closed-loop wake steering control is to leverage existing computationally efficient steady-state wake models and to invoke quasi-stationarity in the flow statistics (see e.g. very recent paper by Doekemeijer et al. (2020), [2]). While the test case in Part 1 explicitly allows for a well-defined assumption of quasi-statistical stationarity, Part 2 will investigate this assumption's utility in a realistic transient ABL state.
Future work should also focus on the synthesis and validation of computationally efficient, transient models for wind farm power prediction.

**Comment #5**
Several question arise from Section 4:
- Is uniform inflow a good and representative case? Does it truly provide validation of optimization, because it turns the turbine in the right direction?
- Line 285 states that AD is good for far wake. But is 4D far wake in uniform inflow? A quick back-of-the-envelope assessment for where the far wakestart can be done using the engineering expression in Sørensen et al., 2015. For a 0.01% inflow turbulence and propagation velocity of 0.7, I get an estimate of 13-14D downstream, before it is fully transitioned to far wake with a Gaussian velocity deficit, which is another of the model assumptions. What is the implication of this assumption?
- Please rephrase for clarification line 377: "reduced in size and intensity". What is reduced in size and what intensity? The deficit? It would be difficult(=impossible) to see from an instantaneous velocity contour. Similarly, it is not possible to see how the figure "indicating that

for larger columns of turbines the potential for power increases due to wake steering are larger" due to the dynamics of the instantaneous flow.

**Response**

Thank you for this comment. While not physical, uniform inflow (zero freestream turbulence) represents a helpful, well-defined case for algorithmic testing and code validation. As the referee has noted, due to the small turbulence mixing in the wake shear layer, the top-hat profile wake will remain further downstream than in a realistic wind turbine wake. Therefore, this example is not a precise representative case where the assumptions of the wake model are exactly matched.

We will rephrase line 377 for clarity and improve the quality of the figures.

To reduce confusion of this section with physical results (atmospheric boundary layer simulations) we will move this section to the Appendix in the revised manuscript.

**Comment #6**

Quantification. Several issues relating to quantification could be improved and clarified.

- Are the power productions in Table 1 normally distribution? Otherwise, the standard deviation could be biased and misleading.

- The inclusion of whether results are significant can only be applauded. Line 434 defines "significant superior...if the mean array power production ....is more than one standard deviation larger than the other". However, the use is biased, because the analysis is essentially only one-sided. The authors only compare controlled mean power - 1 standard deviation to the meanpower of the baseline. The authors should also include the standard deviation of the baseline, because it's two overlapping distributions. That essentially means none of the cases are significantly superior if using the authors definition of statistical significance.

**Response**

The Figure 1 shows the probability distribution of the six turbine array over the control update steps (from LES case ND2). The sum of the six wind turbine array power production is approximately normal as a function of control update steps. Given the sample data deviation from an exact Gaussian distribution, we will use two statistical tests to check the statistical significance of the LES power production data below.

[Figure]

**Figure 1:** Probability distribution over the control update steps for the sum of the six turbine array power production from LES Case ND2.

While the total turbine array power production is approximately normal as a function of the control update steps, we will demonstrate two statistical tests here to show statistical significance for Cases NA and ND2 by assuming normal distributions or not. The null hypothesis is that the two temporal signals of the sum of the six turbine power production are random samples from the same underlying probability distribution.

**Two-sample T-test  (assumes normal distributions):**
P = 6.13E-4, and therefore the null hypothesis is rejected at a 5% significance level.

**Two-sample Kolmogorov-Smirnov test (does not assume normal distributions):**
P = 1.16E-4, and therefore the null hypothesis is rejected at a 5% significance level.

Therefore, the differences between Cases NA and ND2 are statistically significant at a 5% significance level. We will update the manuscript with these further statistical tests for all cases.

**Comment #7(a)**
Zong and Porté-Agel also shows the secondary steering effect, which other recent articles are also investigating, see e.g. King et al., 2020. The secondary steering effect essentially implies that the next wake will also be partly steered for "free"(without power loss). The Liew et al. 2020 reference states that McKay et al. 2013, Bartl et al. 2018, and Hulsman et al.2020, all show that the local inflow direction changes, andthe waked (2nd) turbine should adjust to the inflow. Hence, it would indicate that if the second turbine also yaws 20deg, it actually corresponds to a

larger relative yaw angle. So why would the optimization give the same(or relatively larger) yaw angles for the first 3 turbines?

**Response**

As demonstrated by the recent studies the referee has discussed, secondary steering is an important aspect of wake steering control. Modeling efforts to capture secondary steering are on-going (King et al. 2020 in review in *WES*) or very recently published (Zong and Porte-Agel (2020), which was published after this manuscript was submitted). The wake model methodology used in this study's simulations did not model secondary steering and therefore this effect will not be directly reflected in the model-optimal yaw angles, although the authors note that the lifting line model with EnKF estimation predicts a generally decreasing trend of yaw misalignment values deeper into the array of turbines (e.g. Figure 8). Future work should examine the efficacy of secondary steering models in closed-loop wake steering control, but is out of the scope of the present study since all LES cases would need to be re-run.

**Comment #7(b)**

Should the yaw angles not be reduced at least by the secondary effects or is this not capture by the LES?

**Response**

The yaw angles reported in this manuscript are the local turbine relative yaw misalignment values (i.e. the yaw misalignment of the turbine with respect to its local wind direction measurement), and therefore the secondary steering effect is accounted for in the reporting of the yaw misalignment values from LES but not in the model optimization as discussed in the response to **Comment #7(a)**.

**Comment #7(c)**

A quick estimate of the power yaw loss with PP = 3, yields the following table (see table in review). That means the first turbine yawing 15-20deg looses approx. 10-17% of the power compared to its own baseline(also shown in Figure 16). Line 480 states "power production penalty...is significant beyond 40deg". The 40deg seems "random", as one could just as well argue that losses of 10-17% are significant. In order for the entire farm to produce more, this loss should be recovered by the next turbine. However, the next 3 turbines experience similar losses. Therefore, increase in velocity have to compensate for all these turbine losses. If I use estimates from Figure 16, I read an initial power production of approx. 35% and an optimal power production of approx. 50% for the second turbine relative to the first. Hard to see from this rough estimate if the second turbine actually gain more than what the first turbine losses. Archer and Vasel-Be-Hagh also concluded there needs to be at least 2 turbines downstream a yawed turbine to regain the power loss.

**Response**

Thank you for this comment and for your analysis. We will remove Line 480 for clarity.
Figure 16 only plots the power for the first two turbines in the 6 turbine array (to reduce clutter in the Figure). Since it only plots the first two turbines, it should be noted that the gains at turbine 2 do not directly compensate the losses as turbine 1, since the objective is to maximize

the 6 turbine array, not focus on two-turbine sub-optimization problems which reduces global array optimality.

The power production for turbine 2 shown in Figure 16 accounts for the total power, i.e. the power production of turbine 2 accounts for power reduction due to its own yaw misalignment as well as power increases due to turbine 1's yaw misalignment. The largest relative gains in the 6 turbine array will be the turbines further downwind since their relative yaw misalignment is lower (see Figure 14). The total power production for the 6 turbine array is compared to baseline control in Figure 11.

As the referee correctly notes, the conclusion of Archer and Vasel-Be-Hagh (and Howland et al. (2020)) has motivated the use of the 6 turbine array to test closed-loop wake steering in this study rather than a 2 turbine array.

**Comment #7(d)**

In order to recover the power loss, the wake velocity have to increase by $V2/V1 = (0.50/0.35 * CP0/CPyaw)^{(1/3)}$ where CP0 corresponds to my previous table of 0degyaw, and CPyaw is the remaining. Hence, the velocity increase(V2/V1) on the second turbine needs to be (see table in review). So a 17-20% increase in mean velocity. It would be nice to see some kind of velocity plots, either contour plots before and after control to verify this. Therefore, it would also be nice to see the actual power production of the individual turbines compared to baseline for the various scenarios, because it's difficult to assess which turbines actually make up for the losses and increase the total production in the accumulated plots, e.g. Figure 9. The authors have attempted to do so in Figure 21, but it's very difficult to distinguish. It seems turbines 2, 4 and 5 recover the losses for PP = 3? Perhaps use bar plots with clearer standard deviations?

**Response**

We will add further plots, tables, and discussion to clarify the quantitative results in these simulations. To address the referee's specific concern, we will articulate some quantitative results here, and include a thorough discussion in the revised manuscript.

Focusing on Case ND2, the power production for each turbine is normalized by the baseline control power production (of the same turbine) and is averaged over the control update steps. The resulting normalized power production values are:

| Turbine | Power (normalized by baseline greedy control and averaged over control steps) |
|---|---|
| 1 | 0.844 |
| 2 | 1.102 |
| 3 | 1.182 |
| 4 | 1.159 |
| 5 | 1.192 |
| 6 | 1.106 |

As can be seen from the power production values for Case ND2 (normalized by Case NA), turbine 1 loses approximately 15% of its power production but each downwind turbine generates 10-20% higher power production than baseline control. Since turbine 1 generates larger power, its losses influence the array sum more than the gain of the downwind turbines, but collectively, the average array power production increases 4.6% over greedy baseline

control in this case. As can be seen, with more turbines downwind, the potential for wake steering to increase relative power production over greedy control also increases.

**Minor comments**

**Minor comment #1**
The inclusion of FLORIS (Section 5.4) seems superfluous. I suggest to leave it out to keep the analysis more focused, and shorter.

**Response**
The authors do not agree that Section 5.4 is superfluous. This section tests the influence of the state estimation methodology and FLORIS is a commonly used wake model whose parameters have been empirically defined such that model-based parameter estimation is not used. This comparison shows that without model parameter estimation the wake model does not fit the baseline power production, and as such, _predicts_ power given a yaw misalignment strategy with 5x larger error (Figure 21(b)).

**Minor comment #2**
- The definition of 0deg north seems redundant. Where is it used? It seems to be in conflict withFigure 6, as the wind direction with 0deg north would not be 16deg, but a 254deg.

**Response**
Thank you for this comment, we will modify the manuscript to clarify the terminology.

**Minor comment #3**
Terminology might change over time and for different researchers. Some use kidneyshaped, others curled wake. However, the vortex pair has been observed in the wake behind a yawed turbine prior to 2016 (line 131). The earliest reference I could find was Mikkelsen, 2004, see Section 8.3.

**Response**
We will add the reference to Mikkelsen 2004.

**Minor comment #4**
- Please introduce/explain Table 1 more when it is first presented.

**Response**
We will modify the manuscript accordingly.

**Minor comment #4**
Please provide a proper description of the turbine. Line 398: "selections based on NREL 5MW" does not seem reproducible by other scientists.

**Response**

The reference to the NREL 5 MW turbine was to clarify the selection of the turbine diameter. All relevant computational parameters of the turbine are detailed in Section 3.

**Minor comment #5**
Details are difficult to see in many of the figures. For instance, reduce the axis limits on Figure 15 and similar. Figure 21 is very hard to distinguish.

**Response**
Thank you for this comment. We will modify the figures accordingly to improve visibility.

**References**

[1] Doekemeijer, Bart M., Jan-Willem Van Wingerden, and Paul A. Fleming. "A tutorial on the synthesis and validation of a closed-loop wind farm controller using a steady-state surrogate model." *2019 American Control Conference (ACC)*. IEEE, 2019.

[2] Doekemeijer, Bart M., Daan van der Hoek, and Jan-Willem van Wingerden. "Closed-loop model-based wind farm control using FLORIS under time-varying inflow conditions." *Renewable Energy* (2020).

[3] Ciri, Umberto, Stefano Leonardi, and Mario A. Rotea. "Evaluation of log-of-power extremum seeking control for wind turbines using large eddy simulations." *Wind Energy* 22.7 (2019): 992-1002.

[4] Shapiro, Carl R., et al. "A Wake Modeling Paradigm for Wind Farm Design and Control." *Energies* 12.15 (2019): 2956.

[5] Doekemeijer, Bart M., et al. "Online model calibration for a simplified LES model in pursuit of real-time closed-loop wind farm control." *Wind Energy Science* 3.2 (2018): 749-765.

[6] Shapiro, Carl R., Dennice F. Gayme, and Charles Meneveau. "Modelling yawed wind turbine wakes: a lifting line approach." *Journal of Fluid Mechanics* 841 (2018).

[7] Bastankhah, Majid, and Fernando Porté-Agel. "Experimental and theoretical study of wind turbine wakes in yawed conditions." *Journal of Fluid Mechanics* 806 (2016): 506-541.

[8] Liew, Jaime Yikon, Albert M. Urbán, and Søren Juhl Andersen. "Analytical model for the power-yaw sensitivity of wind turbines operating in full wake." *Wind Energy Science* 5.1 (2020): 427-437.

[9] Medici, D.: Experimental studies of wind turbine wakes: power op-timisation and meandering, PhD thesis, KTH, Stockholm, Sweden, 2005

[10] Fleming, P., Gebraad, P., van Wingerden, J.-W., Lee, S., Church-field, M., Scholbrock, A., Michalakes, J., Johnson, K., and Moriarty, P.: SOWFA Super-Controller: A High-Fidelity Tool for Evaluating Wind Plant Control Approaches, Tech. rep., NationalRenewable Energy Lab. (NREL), Golden, CO, USA, 201

---

## Referee Comment (RC2) · Anonymous Referee #2 · 21 May 2020

Thank you very much for this paper. On the whole I found it to be both well written and thoroughly researched. I also commend the author's on excellently situating this work within the public research and providing a thorough review of the relevant background.

Additionally, I agree with a main suggestion of the paper, that wake steering has reached a point where it is important to not only demonstrate that it can work, but to struggle with the real-time implementation and controls issues. I believe the method proposed in this paper makes a lot of sense, using state estimation to calibrate the wake model. I believe the paper can be accepted with minor revision.

General Comments:

[Figure]

One overall comment I had was I wasn't quite positive of the main, conclusion of the paper. Is that the proposed method is at least successful in the provided tests, or that it is necessary to use a method like this? The paper is a little bit long, I wondered if parts might be condensed also to make more clear what are the bigger, more important, findings? The authors however can freely disregard this suggestion, but I hope this impression is useful.

What is the model of yaw control within the turbine used? Is the turbine free to yaw at any moment? Most turbines have a built-in yaw control strategy which includes some dead-band about vane angle, and an intentionally delayed response to changes in wind direction. If this was not used, how would it change results if it were?

One general comment I had on the sections related to the cos pP parameter, is that the discussions and conclusions are proposed in absolute terms, where relative would be more appropriate. As i understand, pP=3 is "correct" in this LES simulation, and so 2 is 2/3 of correct, and 4 is 4/3 of correct. Most of the numbers are baselined to a correct value of 3, but should be scaled in other simulations or on physical turbines. My main point is to avoid stating that 2,3 or 4 is better/worse and more under-predicting pP by x% leads to, while over-prediciting pP by x% leads to ... In a similar way this would change the statement "a conservative estimate of pP=4" should be used in cases where it's not yet known could be the more reasonable 125% of the value of the most similar published value (in terms of rated power or rotor size).

Specific comments:

The last sentences of the abstract are somewhat confusing before you read the paper

p 13 "likely enhanced in yaw misalignment" see yaw-added recovery in https://www.wind-energ-sci-discuss.net/wes-2020-3/

P23 "wind speed and direction bins of arbitary size" this seems pretty possible using interpolating functions
P23: "using a neural network for example" these ideas are theoretically possible but practical observation suggests that any method who's parameters are not human intelligible will have obstacles because it will be difficult to make in-field adjustments

---

## Author Comment (AC2) · 29 May 2020

**Response to Referee #2**

We thank the referee for their review and their thoughtful comments. Point-to-point responses can be found below, and the relevant changes will be made to the manuscript during the revised submission stage (https://www.wind-energy-science.net/peer_review/interactive_review_process.html), colored in blue.

**General comments:**

**Comment #1**

One overall comment I had was I wasn't quite positive of the main, conclusion of the paper. Is that the proposed method is at least successful in the provided tests, or that it is necessary to use a method like this? The paper is a little bit long, I wondered if parts might be condensed also to make more clear what are the bigger, more important, findings? The authors however can freely disregard this suggestion, but I hope this impression is useful.

**Response**

Thank you for this helpful comment, we will streamline the discussion in the paper to highlight the main contributions.

The main technical contribution of this paper is the development of a closed-loop wake steering methodology for application in transient ABL flows which does not rely on an open-loop offline yaw misalignment lookup table calculation.

The two main conclusions are: 1) that the wake model, when combined with a state estimation approach, is able to predict the power production for the wind farm in yaw misalignment with significantly less error than the standard approach of an empirically pre-calibrated wake model and 2) model parameter uncertainties, such as the uncertainty in Pp, can inhibit the success of a wake steering application and these uncertainties must be carefully accounted for.

**Comment #2**

What is the model of yaw control within the turbine used? Is the turbine free to yaw at any moment? Most turbines have a built-in yaw control strategy which includes some dead-band about vane angle, and an intentionally delayed response to changes in wind direction. If this was not used, how would it change results if it were?

**Response**

The referee is correct that most utility-scale turbines have a native yaw control system which acts outside a deadband of 5-10 degrees based on low-pass filtered wind vane angle measurements. The yaw control strategy used in the present study low-pass filters the wind direction measured at the rotor based on a predefined time constant, leverages the computed turbine-specific wind direction to implement the desired yaw misalignment, and holds the imposed nacelle position for one period of the predefined time constant. This was described on Page 11 Line 315 and will be further expanded on in the revision to improve clarity.

Depending on the dynamics of the ABL, the method we have employed may lead to different results than the method the referee has mentioned. However, in the present conventionally neutral ABL large eddy simulations, the variations in the turbine-specific wind directions as a

function of time due to turbulence and inertial oscillations are relatively small (a few degrees) and therefore the yaw control method we have used and the deadband method will perform very similarly.

For Part 2 of this work, where the closed-loop controller is tested in the transient diurnal cycle, this native yaw control system will likely have a larger impact and we will test various native yaw control strategies.

**Comment #3**
One general comment I had on the sections related to the cos pP parameter, is that the discussions and conclusions are proposed in absolute terms, where relative would be more appropriate. As i understand, pP=3 is "correct" in this LES simulation, and so 2 is 2/3 of correct, and 4 is 4/3 of correct. Most of the numbers are baselined to a correct value of 3, but should be scaled in other simulations or on physical turbines. My main point is to avoid stating that 2,3 or 4 is better/worse and more under-predicting pP by x% leads to, while over-predicting pP by x% leads to ... In a similar way this would change the statement "a conservative estimate of pP=4" should be used in cases where it's not yet known could be the more reasonable 125% of the value of the most similar published value (in terms of rated power or rotor size).

**Response**
Thank you for this comment. The referee is correct that the reference to conservative values of Pp are relative since they depend on the turbine-specific correct, or maximum likelihood estimate, value of Pp. The authors suggest that given a confidence interval on Pp for a given wind turbine, a conservative estimate should be selected for Pp for the calculation of the yaw misalignment strategy since the underestimate of Pp leads to power production loss during wake steering. We will modify the manuscript to consider your comment and discuss the Pp sensitivity in relative terms.

**Specific comments:**

**Comment #1**
The last sentences of the abstract are somewhat confusing before you read the paper
**Response**
We will modify the last sentence of the abstract for improved clarity.

**Comment #2**
p 13 "likely enhanced in yaw misalignment" see yaw-added recovery in[https://www.wind-energ-sci-discuss.net/wes-2020-3/](https://www.wind-energ-sci-discuss.net/wes-2020-3/)

**Response**
Thank you for highlighting this paper, we will add the reference to the yaw-added recovery.

**Comment #3**
P23 "wind speed and direction bins of arbitary size" this seems pretty possible using interpolating functions

**Response**
Thank you for this interesting comment, interpolating functions would be an excellent candidate to use in open-loop lookup table computation.

**Comment #4**
P23: "using a neural network for example" these ideas are theoretically possible but practical observation suggests that any method who's parameters are not human intelligible will have obstacles because it will be difficult to make in-field adjustments

**Response**
The authors agree that methods which are based on first principles or physical phenomena are likely to perform better in a complicated field environment and we leave machine learning questions for future experimentation and improvement.

---

## Editor Comment (EC1) · Katherine Dykes (Editor) · 10 Jun 2020

This review was submitted prior to the deadline but after the system closed.

[Figure]

**Comments to authors**

The authors present their manuscript 'Optimal closed-loop wake steering, Part 1: Conventionally neutral atmospheric boundary layer conditions' in which they discuss their framework for power-maximizing wind-farm control through wake steering. They combine a lifting line wake model with a ensemble Kalman filtered state estimation tuning the model parameters to SCADA data from a virtual wind farm in the form of a large-eddy simulation. They apply their methodology to a uniform-inflow two-turbine test case and a wind farm submerged in a conventionally-neutral ABL. For the latter case, they perform a series of sensitivity tests to investigate the effect of some design choices of the control framework, which is claimed to be the overall goal of the current study.

The research is original, interesting, and holds merit for the overall wind-farm control community. However, I believe the quality of the paper could be significantly improved by taking into account the following comments.

Major comments

1. I believe that the paper could be significantly shortened in some areas, which would highly increase the readability and allow the key messages to be conveyed more clearly. Some examples:

   - The introduction could be considerably reduced without harming its quality: the general introduction in wind energy (up until line approx. line 27) can be omitted, the discussion on induction control could be reduced to simply mentioning that dynamic control is much more promising than static (with some key references, i.e. Annoni et al, Campagnolo et al, Munters & Meyers, Frederik et al.)

   - The literature review at the beginning of Section 2.2 can be shortened

   - Section 2.4 basically discusses a straightforward time lag based on Taylor's hypothesis. This could be significantly shortened.

   - Section 2.5 takes up quite a lot of space with again a detailed review, but very little is said related to the current manuscript, other than 'the update frequency is selected according to the dynamics of the problem studied'. Further, 'Comments on the update frequency are made in Section 5.' (l. 270), hinting on a study where the sensitivity to this frequency is analyzed, where is this exactly? Or do the authors refer to the part where a dynamic approach is compared to a lookup table (i.e. Section 5.1)? In the latter case, please rephrase (l. 270) more exactly.

2. I found the elaboration of the ensemble Kalman filter state estimation algorithm somewhat hard to follow.

   (a) It would be illustrative if the authors could provide a schematic which shows inputs, outputs, and operations of the algorithm. This could

**Fig. 1.**

---

## Author Comment (AC3) · 13 Jun 2020

**Response to Referee #3**

We thank the referee for their review and their thoughtful comments. Point-to-point responses can be found below, and the relevant changes have been made to the manuscript.

**General comments:**

**Comment #1(a)**

The introduction could be considerably reduced without harming its quality: the general introduction in wind energy (up until line approx. line 27) can be omitted, the discussion on induction control could be reduced to simply mentioning that dynamic control is much more promising than static (with some key references, i.e. Annoni et al, Campagnolo et al, Munters & Meyers, Frederik et al.)

**Response**
We will streamline the introduction review.

**Comment #1(b)**
The literature review at the beginning of Section 2.2 can be shortened

**Response**
We will shorten the literature review for brevity.

**Comment #1(c)**
Section 2.4 basically discusses a straightforward time lag based on Taylor's hypothesis. This could be significantly shortened.

**Response**
We will remove this section and state the time lag based on Taylor's hypothesis in a brief fashion.

**Comment #1(d)**
Section 2.5 takes up quite a lot of space with again a detailed review, but very little is said related to the current manuscript, other than 'the update frequency is selected according to the dynamics of the problem studied'. Further, 'Comments on the update frequency are made in Section 5.' (l. 270), hinting on a study where the sensitivity to this frequency is analyzed, where is this exactly? Or do the authors refer to the part where a dynamic approach is compared to a lookup table (i.e. Section 5.1)? In the latter case, please rephrase (l. 270) more exactly.

**Response**
We will streamline and re-phrase the discussion in Section 2.5 and on Line 270 to improve clarity.

**Comment #2(a)**

It would be illustrative if the authors could provide a schematic which shows inputs, outputs, and operations of the algorithm. This could be similar to figure 1 which shows $\alpha$ and $\gamma$ as inputs to the state estimation, however their role is not discussed in section 2.2.

**Response**
We will include a schematic to describe the state estimation methodology and clarify the role of $\alpha$ and $\gamma$.

**Comment #2(b)**
Further, the statement 'The EnKF is computationally superior ... but this may lead to spurious correlations in the state representation' is somewhat confusing. Does this affect the results in the current paper?

**Response**
We will remove the statement which does not directly relate to the results of the current paper.

**Comment #2(c)**
The EnKF is particularly well-suited for discretized PDE problems. Why? add a suitable reference perhaps

**Response**
We will remove this statement which is not critical to the current paper.

**Comment #2(d)**
The outcome of the EnKF is a vector with k w and $\sigma$ for all turbines except the last one, for which this is not relevant (l 191 p 7). However, in your equations, this last turbine is included in the state estimation, e.g. see Equation (7). Is this correct?

**Response**
Thank you for noting this notation error. The referee is correct that the furthest downwind turbine does not need to be included in the state estimation, and in fact, its parameters are irrelevant for the wake model. We will adjust the state estimation equations accordingly.

**Comment #2(e)**
Equation (18): should all $\pi$'s be replace with $\hat{\pi}$ here? If not, how is $\pi$ defined?

**Response**
Thank you for noting this typographical error in Equation (18). The $\pi$'s should have been $\psi$'s and we have fixed the error.

**Comment #3(a)**
The authors use an actuator disk model. Some more details would be welcome. Is it a rotating or non-rotating model (I assume the latter). The ADM is accurate in the far wake, yet at a turbine spacing of 4D it is doubtful whether far wake conditions are met. Somme comments on

this would be welcome. How does the ADM deal with the dependency of thrust forces and power extraction on yaw angle? Is this a standard cos 2 ( $\gamma$ ) and cos 3 ( $\gamma$ ) respectively? How does this influence conclusions based on the sensitivity of the control framework on P p in section 5.3?

G

**Response**

The actuator disk model used in the large eddy simulations presented in this study does not include rotation, it is a standard actuator disk model. We will include more details on the ADM in the manuscript and the description is also stated here. The referee is correct to note that the ADM has some discrepancy with the rotating ALM in the near-wake region.

To be clear, the relative turbine spacing in the conventionally neutral ABL simulations is between 4-5D in the mean hub-height wind direction since the spacing is 4D in the x-direction and the turbines are aligned at 18 degrees to the x-axis.

The wind speed profile at the wind turbine hub height approximately 4D downwind of the leading turbine, incident on the second turbine, is shown in Figure 1. The wake profile exhibits a Gaussian shape, rather than a top-hat shape, suggesting that with the turbulence intensity and length scales present in the conventionally neutral ABL LES, 4-5D spacing is sufficient to establish far wake behavior with the actuator disk model used in this study. This is consistent with the expected location of far-wake behavior onset (experimentally found in yawed rotating turbines to be approximately x/D=3 [1]).

The goal of the current study is to test the efficacy of the closed-loop control algorithm, and therefore, the focus was not to precisely match the wakes observed in full-scale wind farms through LES; future work should also incorporate more advanced wind turbine models.

[Figure]

**Figure 1**: Time averaged wind speed profile at x/$\delta_0$=3.5 at the wind turbine hub height. The flow at this x-location is incident on the second wind turbine in the array. The wind speed exhibits a Gaussian profile, suggesting the onset of far-wake behavior [see e.g. 1].

Within the actuator disk model formulation, the dot product is taken between the incident wind velocity vector and the wind turbine rotor normal vector to calculate the perpendicular velocity vector. The thrust force normal to the rotor area is computed using the perpendicular velocity, the axial induction factor (a=0.25), and the coefficient of thrust ( $C_T = 0.75$ in the

current study). The forces are then projected into the computational domain coordinate system. In this fashion, the dependence of the thrust force (with thrust being defined as parallel to the mean velocity at hub height) on the yaw misalignment angle is $cos^2(\gamma)$ given uniform inflow conditions. Importantly, with non-uniform flow conditions, the thrust is not guaranteed to follow $cos^2(\gamma)$. The ADM power production is defined as $P = u_p \cdot F_T$ where $u_p$ is the velocity perpendicular to the rotor area and $F_T$ is the thrust force. This results in a power production dependency on the yaw misalignment angle of $cos^3(\gamma)$, again, for uniform inflow conditions only. For heterogeneous flow conditions, the scaling for thrust and power as a function of yaw misalignment is approximately $cos^2(\gamma)$ and $cos^3(\gamma)$, respectively, although it may deviate from these approximate scaling expectations. We will include further comments on the ADM in the revised manuscript.

The particular "ground-truth" value of $P_p$, as discussed in Section 5.3, is wind turbine model specific. The current LES experiments detail the sensitivity of the wind farm power production, and therefore the efficacy of wake steering, to the estimate of $P_p$ for a ground-truth $P_p = 3$, but the sensitivities noted in the current study are not expected to depend on the particular ground-truth value for $P_p$. For example, the current experiments show that an estimate of $P_p = 2$ when the ground-truth value is 3 leads to power production loss compared to standard operation. For a different wind turbine model with a ground-truth of $P_p = 2$, we would then expect an estimate of $P_p = 1.5$ may lead to undesirable results, for example, while $P_p = 2$ would be an excellent estimate. The results of the current study are not absolute, in the sense that $P_p = 4$ is always a good choice, but relative, in the sense that a conservative estimate for $P_p$ is wise given the $P_p$ parameter uncertainty which is present in wake steering applications. We have refined the discussion in Section 5.3 to clarify this point.

**Comment #3(b)**
p.12, l 323 – 325: '... without the influence of variable turbine operation, the flow is identical to machine precision between yaw aligned and yaw misaligned cases'. I would suggest rephrasing this. I understand the authors want to convey that both cases are started from identical initial conditions, and any differences can hence be attributed to differences in farm controls. However, stating that these simulations are 'identical to machine precision' is somewhat deceptive in the simulation of a chaotic dynamical system. In such systems, differences (even at machine precision levels e.g. by adapting compiler optimization levels), will grow exponentially in time, resulting in a completely different turbulent flow realization. I'm not saying this is the case in the current simulations, but a better phrasing would simply be to remove the 'to machine precision' part.

**Response**
Thank you for this comment. The referee is correct that differences in turbulent systems grow exponentially in time and even modifications in the compiler optimization could cause these differences which grow to O(1) errors. We have carefully ensured that these floating-point differences are eliminated by fixing the initialization (to machine precision), compiler optimization, and processor topology in the present study to allow for quantitatively rigorous

comparisons between the various control simulations. We have added a footnote to clarify the points the referee has raised.

We can further demonstrate the point the referee has raised here. We perform a numerical experiment with three simulations. In the conventionally neutral LES flow, we fix all parameters and initializations to machine precision and the control architecture is fixed between the three cases. In two cases (#1 and #2), the processor topology is fixed while in the third case (#3), the compiler optimization changes the processor topology. The power production for Cases #2 and #3 normalized by the power from Case #1 are shown below in Figure 2. Cases #1 and #2 remain in quantitative agreement to machine precision while Cases #1 and #3 diverge. The deviation between Cases #1 and #3 are a consequence of round-off differences that occur due to global reduction (sum) operations with modified processor topology.

All comparison cases in this paper were performed with the comparison methodology enforced in Cases #1 and #2.

[Figure]

**Figure 2:** Processor topology experiment. In the left plot, the wind turbine array power production (Case #2) is normalized by a separate LES case (Case #1) with fixed control architecture and initialization as well as a fixed processor topology. In the right plot, the power (Case #3) is normalized by a separate LES case with fixed control architecture and initialization but a modified processor topology (Case #1). The normalized power from Case #3 oscillates around 1 as a function of time.

**Comment #4(a)**

The relatively low grid resolution, combined with the uniform inflow creates a possibly problematic setup where the wake behind the first turbine is artificially stabilized, resulting in very low baseline power in row 2, and hence huge gains to be obtained from any type of control. A remark could be added to the text on this. This is somewhat mentioned in l. 361, but it would be good to mention that this case is highly dependent on things like grid resolution, SGS model, and hence physical results should be interpreted with care.

**Response**

Thank you for this comment. The referee is correct that LES of uniform inflow impinging on wind turbine models at typical LES resolutions leads to an artificial stabilization as mentioned in the discussion in Section 4 (Line 361). This challenge of turbulent transition in wind turbine model wakes for uniform inflow and the influence of the SGS model was discussed in detail in Howland *et al JRSE* (2016) [2]. As a result of this comment, and those of the other referees, we have moved the uniform inflow test case section to the Appendix to reduce confusion between this unphysical algorithmic test case and the physical conventionally neutral LES results.

**Comment #4(b)**
Figure 3 and Figure 4: x-axis is labeled time step. Is this simulation time step of control update step? I'm suspecting the latter, but it would be better to be explicit, similar to how axes are labeled in later figures.

**Response**
Thank you for noting this typographical error. The x-axes for Figures 3 and 4 should state 'Control update step,' and we have updated the manuscript accordingly.

**Comment #4(c)**
p. 14, l. 369: The authors indicate their fear of an overparametrized model which is overfitting by spurious anti-correlation of k w and $\sigma$ 0 . Would it make sense / be possible to directly try and obtain these parameters from a time-averaged LES flow field and hence quantify the 'correct' parameter values?

**Response**
Thank you for this suggestion. This is an interesting idea and it is possible to quantify an empirical best-fit for the wake spreading rate kw in the LES setting. In fact, this has been the previous wake modeling approach to the wind farm controls problem (see e.g. Niayifar & Porte-Agel (2015) [3] and the FLORIS model). However, this methodology is impractical in a field setting for continuous parameter estimation where nacelle-mounted LiDARs are not often available and *a priori* parameterized values for physical constants can lead to inaccurate power predictions (see comparison with empirical Gaussian wake model in Figure 21). The purpose of the present study is to establish a framework which relies only on readily available SCADA data for model parameter estimation, and therefore we have not focused on flow-field empirical parameter fitting in this study. Future work should investigate the efficacy of physics-based constraints during optimization of the wake model parameters, but this is out of the scope of the current study.

**Comment #4(d)**
Concluding, the overall added value of this section is quite limited in my opinion. The main contribution would be to have a very basic test case, showing that the EnKF and yaw angles are relatively stable for steady flow conditions. Therefore, I believe it would be better if the section is introduced and discussed with this aim.

**Response**

As discussed in the response to **Comment #4(a),** we have moved this section to the Appendix to focus on the physical results in the conventionally neutral ABL setting.

**Comment #5(a)**

Table 1 is introduced early on. However, by itself, the table is insufficiently explained to completely understand which case is which. An example is the NA case, which is not clearly defined in the text until Appendix C. Also, the naming and order in which these cases are presented in the table could be greatly improved. For example, Section 5.1 discusses NL and ND2, whereas ND1 is only introduced in Section 5.2. The naming also gets very confusing later on, e.g. the case with P p = 2 is called ND4 and P p = 4 is called ND5. This results in the reader constantly having to go back to Table 1 to recall which case is which. More logical naming would prevent this, e.g. replace ND4 by NDP2, ND5 by NDP4.

**Response**

We will describe the cases and Table 1 in more detail and earlier in the manuscript to improve clarity and we have re-named the cases according to the referee's suggestions. Note that we have also switched the names of Case ND1 and ND2 in the revision as per the referee's suggestion.

**Comment #5(b)**

The final column in Table 1 is used to determine statistical significance of improvements over the basecase. It would be more illustrative to plot this in a barplot including errorbars, because seeing whether these values overlap for different cases from numerical data is not trivial.

**Response**

Thank you for this suggestion, we have incorporated a barplot (Figure 5) with corresponding errorbars to clarify these data and their statistical significance.

**Comment #5(c)**

Section 5.1: How is the lookup table in the NL case generated? Is this simply by running 1 control window, and then keeping the yaw angles constant? How robust is this? I.e. how would these steady yaw angles differ if they were generated based upon a state estimation from a different time window / turbulent flow realization? This would have an impact on statistical significance of the results.

**Response**

As discussed on Line 416 of the manuscript, the lookup table approach is approximated by: "The lookup table control is approximated by fixing the yaw misalignment angles as a function of time after the initial optimal angles are computed during the first yaw controller update"

Generating the yaw angles in a different time window would lead to small changes in the yaw misalignment values computed by the model-based optimizer as the referee has mentioned. This point was investigated by the authors extensively before submission of the manuscript, although it was not shown for brevity in the already lengthy manuscript. We will articulate the results here and add a brief discussion in the revised manuscript.

While generating yaw misalignment angles in a different time window/turbulent flow realization with slightly different mean wind direction, wind speed, turbulence intensity, etc leads to slightly different values of yaw misalignment, the conclusions of the study are not affected since the changes to the wind farm power production are not statistically significant. This experiment was investigated for Case ND1 and Case ND1R, the robustness version of Case ND1 which is algorithmically identical but initialized at a later LES temporal instance.

The yaw misalignment values implemented in the first control update step of Case ND1 are (from first to last turbine in degrees):
[16.8163, 17.0334, 16.8953, 14.7568, 2.8090, 0.0000]
The increase in power production over baseline control (Case NA) is for Case ND1: 4.59% +/- 2.34%
The yaw misalignment values for Case NL are identical to the first step of Case ND1 by definition.

In a robustness test case, the conventionally neutral ABL LES is run for several more physical hours before the yaw misalignment strategy is switched on the and resulting yaw misalignment values are computed in the separate turbulent time window realization (from first to last turbine in degrees):
[15.5785, 15.8899, 14.4753, 12.1178, 3.7497, 0.0000]
Case ND1 was repeated, starting from the later time, and the increase in power production over baseline control (Case NAR, which was also re-run for the later time window) is for Case ND1R: 5.7% +/- 2.03%
The yaw misalignment values for Case NLR are identical to the first step of Case ND1R by definition. Therefore, changing the time window for the lookup table approximation did not significantly change the yaw misalignment values, as the referee has suggested.

The differences in power production for these two cases are not statistically significant as characterized by a one-sided two-sample Kolmogorov-Smirnov test at a 5% significance level. Further, the yaw misalignment values are qualitatively and quantitatively similar. Therefore, while the specific time wind in which the lookup table (or dynamic yaw) control is implemented has some influence on the quantitative results, we have demonstrated that it does not influence the statistical significance of the results. We have added a discussion in the manuscript to clarify this point.

**Comment #5(d)**
Section 5.1: p. 18, l. 438. a lot of explanation is given for a non-significant performance difference. Does this make sense?

**Response**
We will modify the discussion to highlight that the result is not statistically significant.

**Comment #5(e)**

For figures like 8,9,10,11, it would improve direct interpretability of the figures if the case name was included directly on the plot instead of only in the caption. There is ample whitespace left to do this everywhere.

**Response**
We will add the case numbering to the plots.

**Comment #5(f)**
Section 5.2: p. 22, l. 492 the authors claim 'The most successful dynamic control framework ... is the static state estimation methodology'. Please add the casename here explicitly (ND3 I presume). Based on what metric is ND3 more successful than ND2? Power extraction differences were mentioned not to be significant.

**Response**
As with **Comment #5(d)**, we will limit the differentiation of cases which are not statistically significant differences.

**Comment #5(g)**
Section 5.2: p. 23, l.496: 'This potential dependence of k w and $\sigma$ 0 on yaw misalignment was not incorporated explicitly ...' Is this dependence not implicitly accounted for through the EnKF?

**Response**
This is an excellent observation by the referee. The goal of the data-driven EnKF methodology is that the dependence is implicitly accounted for. The authors were suggesting that future work could also incorporate this dependence explicitly in a physics-based modeling strategy. We will clarify this in the manuscript.

**Comment #5(h)**
Section 5.2 last paragraph: This comparison to NL lookup table seems a bit out of place. Could this be moved to Section 5.1? I understand there are chronological dependencies in how you want to write down observations, but the current narrative is somewhat confusing.

**Response**
Thank you for this suggestion, we have reformulated the narrative to improve clarity.

**Comment #5(i)**
Section 5.3 p.25, l.532: 'Interestingly, ND5 outperforms ND2, but not significantly' This statement is confusing, if its insignificant, then the outperformance is not to be distinguished from statistical noise, so calling that interesting seems contradictory. Please remove or rephrase

**Response**
As with **Comment #5(d),** we will remove discussion of statistically insignificant results.

**Comment #5(j)**

Section 5.4: the part on FLORIS can probably be omitted. Further, this section basically also quantifies a dependence on P p , which was already the subject of Section 5.3. Consider renaming sections to avoid overlap in their definition.

**Response**
The authors believe that a comparison with an empirical physics-based model without state estimation is warranted to show the benefit of the EnKF on power production predictions. The lifting line model has not received an empirical calibration treatment as the Gaussian wake model has [see ref. 2], and therefore FLORIS was selected as a comparison. It is important to note that this comparison is predominantly a commentary on the success of the EnKF for improving wake model state estimation, rather than a comparison between the lifting line and Gaussian wake models.

**Minor/technical comments:**

**Comment #1**
The title of the paper could be improved such that the goal of the research is reflected therein, namely quantifying the sensitivity to design choices in the control framework. Currently, the title 'Optimal closed-loop wake steering' is rather generic, and mentioning that it is 'part 1' of a two-parts paper in my opinion degrades the idea that the current work is self-contained.

**Response**
Thank you for this comment. This article was titled 'Part 1' since there will be a follow-up 'Part 2' which will focus on wake steering in diurnal cycle simulations. Each article will be self-contained, as the present article is.

**Comment #2**
p.3, l.79: mentioning that accurately predicting greedy base-line power production is the main challenge in wake steering control, is somewhat exaggerated in my opinion.

**Response**
We will modify the language in the manuscript to highlight that dynamic wake steering does indeed carry other technical challenges aside from the mentioned baseline power production prediction.

**Comment #3**
The output of your controller is a time-series of yaw angles $\gamma$ (t), do the ADM in the LES directly impose these yaw angles, or is there a limitation on the yaw rates? Figure 3 seems to indicate a very large jump in yaw angle within a single timestep. Is this technically feasible?

**Response**
Yaw control rates are typically ~0.5°/s. The largest jump in yaw misalignment in the present study is from greedy control to wake steering control in 'Control update step 1' of the simulations. This jump has a maximum of 30°, which would take approximately 60 seconds to implement. This time is significantly less than the Taylor's hypothesis time lag (Section 2.4) and therefore this yaw rate will not influence the results. For a dynamic wake steering controller with a more rapid yaw update frequency, this effect should be considered.
This was discussed in the original manuscript on Line 480 but we have moved the discussion to the LES formulation Section 3 to improve clarity.

**Comment #4**
p.7, l. 184: number of turbines N T , l. 192: number of turbines N t (either use N T or N t )

**Response**
We will fix the typographical error.

**Comment #5**
p.9, l.234: I believe the algorithm should be denoted as Adam, not ADAM

**Response**

We will fix the typographical error.

**Comment #6**

p.10,l.280: add the continuity equation to the momentum equation

**Response**

We will add the continuity equation for completeness.

**Comment #7**

p.10,l.264: psuedo vs. pseudo?

**Response**

Pseudo, we will fix the typographical error.

**Comment #8**

Figure 2 and Figure 6 could be improved by adding a snapshot of a velocity field in addition to the purely schematic domain presentation.

**Response**

We will add a velocity field snapshot.

**Comment #9**

p.13,l.345: in incorporated vs is incorporated

**Response**

We will fix the typographical error.

**Comment #10**

p.20,l.471: proportionality constant $\sigma$ 0

**Response**

Thank you, we will adjust the manuscript accordingly.

**Comment #11**

p.21,l.472: Gaussian wake does NOT have a clear trend

**Response**

We will fix the typographical error.

[revised manuscript text omitted]

---

## Author Response (AR2)

**Response to Referee #1**

We thank the referee for their review and their thoughtful comments. Point-to-point responses can be found below, and the relevant changes have been made to the manuscript.

**Comment #1**

The paper is improved significantly, i.e. language, statistical significance, several descriptive figures.

**Response**
Thank you for this comment, and for your thorough reviews which aided the improvement of the paper.

**Comment #2**
Very long still, many appendices. Are they all necessary? Appendix C is completely standard approach, and can be explained with a single (or two) sentence(s). No need to include a page. Appendix E is pretty self-contained that it does not add much to the actual article, because it's a different flow scenario. I get a little concerned when I look at Figure F4. There are clear numerical wiggles present, which presumably stems from their numerical scheme and/or their coarse simulations.

**Response**
In order to shorten the paper, we have removed the uniform inflow Appendix F as per the referee's suggestion. We believe that Appendix C is a necessary discussion to ensure the reproducibility of the results, but we have shortened the discussion as per the referee's comment.

**Comment #3**
What does lines and dots refer to in Figure 19?

**Response**
Thank you for observing the missing legend. The lines are the LES power production results and the dots refer to the wake model estimated power. The legend has been added to Figures 16 and 19.

**Comment #4**
Their line 705 in the conclusion is long and does not read very well, although it is very important.

**Response**
We have modified the discussion in line 705 in the conclusion to improve clarity.

**Comment #5**

The assumption on how CT changes with yaw angles affects all wakes. How can the authors argue that these can be investigated independently, and that the model will simply compensate for any error? Compensating makes it unphysical. It's mentioned as a point for future work but a stronger discussion is warranted of how drawing conclusions from the current work is limited by this.

**Response**

The referee is correct in their observation that compensating for an incorrect estimate of $C_T(\gamma)$ would be unphysical, we apologize for the confusion in our response, as we were not suggesting that would be an acceptable model compensation.

The actuator disk model (ADM) used in the present LES simulations enforces $C_T(\gamma) \approx C_T(\gamma = 0) \cdot cos^2(\gamma)$, and therefore, $C_T(\gamma)$ was treated as a known quantity in this study. Deviations from $cos^2(\gamma)$ arise due to non-uniform inflow conditions. In uniform inflow, $C_T(\gamma) = C_T(\gamma = 0) \cdot cos^2(\gamma)$ with the presently used ADM.

In general, $C_T(\gamma)$ is not directly known, for a given turbine, and may not follow the actuator disk $cos^2(\gamma)$ form. Therefore, for the wake model, $C_T(\gamma)$ is an unknown parameter and introduces uncertainty into the wake model estimation and yaw optimization. The authors suggest that considering $C_T(\gamma)$ uncertainty in the wake model would also be a useful exercise, as was the wake model $C_P(\gamma)$ (or $P_P$) uncertainty considered in this study.

**Comment #6**

It is not true that aero-elastic codes automatically gives Pp = 3. For example, studies done using one aeroelastic code found 1.45 standalone and 1.75 in when coupled to an LES code.

**Response**

The referee is correct that aero-elastic codes do not automatically give $P_P = 3$, although the standard assumptions used in blade element momentum applications, including uniform inflow and $\lambda(\gamma) \sim cos(\gamma)$ (where $\lambda$ is the tip-speed ratio operating point) do often lead to a prediction of $P_P = 3$. As the referee has stated, this often requires LES to more reliably predict $P_P$ (see also discussion by Fleming *et al.* (2017) [1]).

The specific value of $P_P$ depends strongly on the incident wind conditions and the turbine control system, and therefore, as the referee states, it is certainly possible to have $P_P \neq 3$ in aero-elastic simulations, depending on these inputs. These dependencies were recently considered by the authors in a separate study [2]. However, the nonlinear impact of the control system and the incident wind conditions are challenging to predict *a priori*, and therefore, the authors believe the consideration of $P_P$ uncertainty is informative and useful to the wake steering community.

**Comment #7**

The transient/dynamics in these particular results are overemphasized. It reads as if it is building towards a "part 2". The results presented here are not as dynamic as they make them appear. Usually, articles should be self-contained and not directed to some future publication.

**Response**

Transience, or deviations from the quasi-statistically stationary state of conventionally neutral ABL turbulence, arises in the conventionally neutral ABL due to inertial oscillations which are slowly evolving on Coriolis timescales. The slowly evolving conventionally neutral ABL provides a useful benchmark for wake steering control algorithmic development, while still providing ABL turbulence, Coriolis effects, capping inversions, and other key features of the ABL.

**Comment #8**
The secondary wake steering should be present in the LES. What is the baseline? This should be made very clear in the work. i.e. can the turbines yaw freely to align with the local wind direction in the NA simulation? It's difficult to understand from line 426-428 if it is nor see it from Figure 7. If the turbines does not include an actual yaw controller, the baseline could be off, and hence the results may be biased. See previously mentioned references and address this issue more clearly.

**Response**
The secondary steering effect is present in the LES. As discussed on line 690 of the previous version and on Line 298 of the revised submission, the baseline control system aligns each turbine with its local wind direction measurement. Therefore, the baseline control does include a yaw controller, as discussed in the manuscript. We have added an additional sentence in the manuscript to ensure this point is clear.

[revised manuscript text omitted]